# ScaleDiff: Higher-Resolution Image Synthesis via Efficient and Model-Agnostic Diffusion

**Sungho Koh**
Hanyang University
ksh000906@hanyang.ac.kr

**SeungJu Cha**
Hanyang University
sju9020@hanyang.ac.kr

**Hyunwoo Oh**
Hanyang University
komjii@hanyang.ac.kr

**Kwanyoung Lee**
Hanyang University
mobled37@hanyang.ac.kr

**Dong-Jin Kim**[*]
Hanyang University
djdkim@hanyang.ac.kr

## Abstract

Text-to-image diffusion models often exhibit degraded performance when generating images beyond their training resolution. Recent training-free methods can mitigate this limitation, but they often require substantial computation or are incompatible with recent Diffusion Transformer models. In this paper, we propose **ScaleDiff**, a model-agnostic and highly efficient framework for extending the resolution of pretrained diffusion models without any additional training. A core component of our framework is Neighborhood Patch Attention (NPA), an efficient mechanism that reduces computational redundancy in the self-attention layer with *non-overlapping* patches. We integrate NPA into an SDEdit pipeline and introduce Latent Frequency Mixing (LFM) to better generate fine details. Furthermore, we apply Structure Guidance to enhance global structure during the denoising process. Experimental results demonstrate that ScaleDiff achieves state-of-the-art performance among training-free methods in terms of both image quality and inference speed on both U-Net and Diffusion Transformer architectures.

## 1 Introduction

Diffusion models have recently emerged as the leading approach in image generation [7], demonstrating the ability to synthesize high-fidelity images from simple text prompts [1, 3, 9, 21, 30]. While these models achieve impressive results at standard resolutions (e.g., under $1024^2$), their performance significantly degrades when generating images at higher resolutions (e.g., beyond $2048^2$), often producing artifacts such as repetitive patterns and structural distortions [12, 19]. However, training diffusion models directly on higher-resolution datasets is prohibitively expensive, requiring both large-scale, high-quality data and substantial computational resources.

As a result, recent research has focused on extending pre-trained diffusion models to generate higher-resolution images in a training-free manner [2, 8, 12, 16, 17, 19, 20, 22, 23, 49]. However, most of the existing methods are primarily designed for U-Net-based models [1, 30], and we observe that many existing methods are inapplicable [12, 16] or exhibit limited effectiveness [20, 28] when applied to recent Diffusion Transformer (DiT) models [9, 10, 21, 29]. Figure 1 highlights this issue, showing clear qualitative differences when existing methods are applied to DiT models. Although patch-based methods [2, 8, 22, 23] such as MultiDiffusion [2] are inherently architecture-agnostic and can generate detailed results with DiT models by processing the image in patches at its original trained resolution, they require significant computational redundancy to process overlapping patches. This

---

[*]Corresponding author.

39th Conference on Neural Information Processing Systems (NeurIPS 2025).

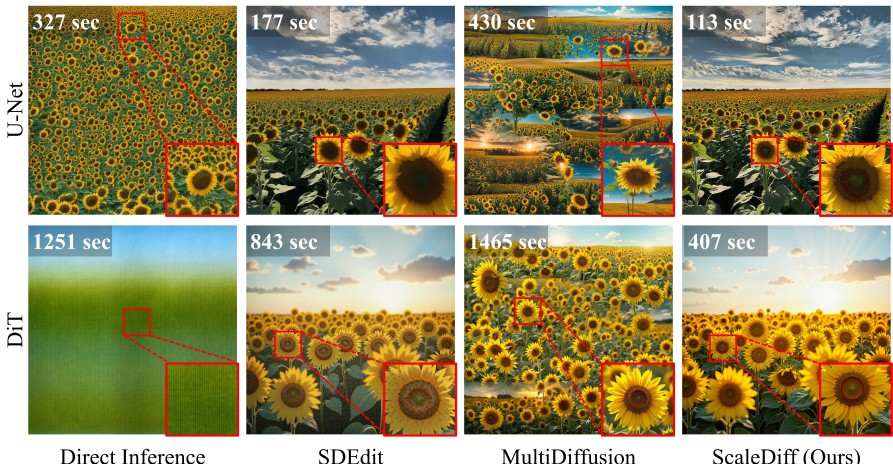

Figure 1: **Comparison between U-Net (SDXL) and DiT (FLUX).** Zoom in for a better view. Elapsed time to generate the image is shown in the top-left corner. Images are generated at $4096^2$.

inefficiency creates a major bottleneck for scalable higher-resolution image synthesis in real-world applications, highlighting the need for more efficient and architecture-agnostic solutions.

In this work, we propose **ScaleDiff**, a highly efficient and model-agnostic framework for extending the resolution capability of pre-trained diffusion models without any additional training. In particular, we introduce **Neighborhood Patch Attention (NPA)** to address the computational redundancy inherent in conventional patch-based methods. In self-attention layers, NPA divides the queries into *non-overlapping* patches and computes attention individually using key and value patches gathered from overlapping spatial neighborhoods. For non-self-attention layers (e.g., MLP), which are less sensitive to resolution, we process the full tensor directly. This design eliminates duplicate computations caused by overlapping image regions, ensuring seamless transitions across patch boundaries. We leverage an iterative upsample–diffuse–denoise pipeline [28] to generate higher-resolution images with global semantic coherence. While prior works [20, 44] perform upsampling in RGB-space, this often results in oversmoothed outputs and a loss of fine details [18, 20]. To address this, we introduce **Latent Frequency Mixing (LFM)**, which refines the RGB-space upsampled latent by replacing its low-frequency components with those from an alternative upsampling path in latent space. Finally, to further enforce global consistency, we incorporate Structure Guidance (SG) [17, 20, 39]. Unlike previous approaches that operate in RGB-space [20], our method applies SG in the latent space to avoid unnecessary computational overhead. SG reinforces structural coherence by aligning the low-frequency components of the model's intermediate prediction with those from a reference latent.

Our main contributions are summarized as follows: (1) We present **ScaleDiff**, a model-agnostic framework demonstrating state-of-the-art results among training-free methods for higher-resolution image generation, achieving significant improvements in inference speed on both U-Net and DiT models. (2) We propose **NPA**, an efficient attention mechanism that reduces computational redundancy by using *non-overlapping* patches in self-attention layers. (3) We introduce **LFM**, a technique integrated with SG to enhance global structural coherence and fine detail synthesis during the denoising process.

## 2   Related Work

**Text-to-Image Generation.** Text-to-image generation has advanced rapidly with the development of diffusion models [15, 40, 41], which generate images by progressively denoising random noise. A major driver of this progress has been the integration of powerful text encoders—most notably CLIP [32]—which enable conditioning image generation on natural language prompts. Early methods such as DALL-E [3] and Imagen [36] demonstrated the potential of large-scale language-vision alignment. The introduction of the Latent Diffusion Model (LDM) [34], further improved efficiency by conducting the diffusion process in a lower-dimensional latent space, facilitating practical high-resolution synthesis. Recent works [5, 6, 9, 10, 21] continue to push the boundaries, exploring

alternative generative frameworks like Rectified Flow [26] and architectural innovations such as Diffusion Transformers (DiT) [29], which replace traditional U-Net [35] backbones with transformer-based architectures [45] to enhance scalability and performance.

**Higher Resolution Image Generation.** Scaling diffusion models to high resolutions often results in repetitive artifacts and structural distortions when naively extrapolated beyond their training resolution [12]. Several methods [6, 11, 25, 33, 43, 47] address this by fine-tuning or training on high-resolution datasets. Despite these efforts, their scalability remains limited due to the fundamental scarcity of high-resolution data and the sharply increasing training cost with image size.

To mitigate these challenges, recent work explores training-free strategies [2, 8, 12, 16, 17, 19, 20, 22, 23, 49] that extend the pretrained model's resolution methods. [2, 8, 22, 23] subdivide the target high-resolution images into overlapping trained-resolution patches, which are processed individually and then stitched together. However, it significantly increases computation due to the necessary overlap and suffers from object repetition issues. Another line of research [12, 16, 19, 49] alters the internal behavior of the model during inference. For example, ScaleCrafter [12] introduces dilated convolutions into the U-Net to expand its receptive field and reduce repetition. However, these modifications are often architecture-specific and tend to degrade image quality at ultra-high resolutions. Editing-based pipelines [17, 20] generate an image at the model's native resolution, upsample it, and then refine it using techniques such as SDEdit [28]. Nevertheless, these editing methods rely on the base model's ability to generate strong local details at higher resolutions—a task that U-Net models can typically handle, but DiT models often struggle with. Compared to prior works, ScaleDiff generates high-resolution images with fine details regardless of the underlying model architecture, while significantly reducing computational overhead.

# 3 Methods

Given a diffusion model trained on fixed-resolution latents $z \in \mathbb{R}^{h \times w \times d}$, our goal is to generate higher-resolution image latents $Z \in \mathbb{R}^{sh \times sw \times d}$, where $s \geq 1$ denotes the scaling factor. To achieve this, we propose ScaleDiff, a training-free and model-agnostic framework that efficiently extends pre-trained diffusion models to higher resolutions. ScaleDiff consists of two main components. First, we introduce Neighborhood Patch Attention (NPA) (Section 3.2), an efficient attention mechanism applicable to both U-Net and DiT architectures that enables the processing of higher-resolution latents. Second, we present the ScaleDiff Upscaling Pipeline (Section 3.3), which builds upon the SDEdit framework [28]. The pipeline incorporates two key techniques: (i) Latent Frequency Mixing (LFM), which refines the reference latent to enhance details, and (ii) Structure Guidance (SG), which maintains global consistency by aligning the low-frequency components of intermediate latent predictions with those of the refined reference.

## 3.1 Backgrounds

**Latent Diffusion Model.** Latent diffusion models [29, 34] first compress an input image in RGB-space into a lower-dimensional latent $z_0 \in \mathbb{R}^{h \times w \times d}$ via an encoder $\mathcal{E}$. This enables subsequent diffusion and denoising to be performed more efficiently in latent space rather than directly on high-resolution pixels. During training, Gaussian noise is gradually added to the clean latent $z_0$ from $t = 0$ to $T$, following the forward process:

$$q(z_t|z_0) = \mathcal{N}(z_t; \sqrt{\bar{\alpha}_t}z_0, (1 - \bar{\alpha}_t)\mathbf{I}), \tag{1}$$

where $\{\bar{\alpha}_t\}_{t=0}^{T}$ is a set of prescribed noise schedules. A denoising network, often based on a U-Net [35] or Transformer [45] architecture, is trained to predict the noise added to $z_t$. During inference, sampling starts from random latent $z_T \sim \mathcal{N}(0, \mathbf{I})$. The trained network iteratively predicts the noise and denoises $z_t$ to estimate $z_{t-1}$, progressively refining the latent until the final clean representation $z_0$ is obtained. This $z_0$ is then decoded to the pixel space by a decoder $\mathcal{D}$ producing the final image.

Self-attention is key to capturing global context in diffusion models as it allows each token to weigh its interaction with all other tokens. The self-attention output $O$ is formulated as

$$O = \text{softmax}\left(\frac{QK^T}{\sqrt{d}}\right)V, \tag{2}$$

where $Q, K, V$ are the Query, Key, and Value matrices derived by linearly transforming the features extracted from $z_t$ through the network. Especially in transformer architecture, $Q, K, V \in \mathbb{R}^{h \times w \times d}$,

sharing the same spatial dimension as $z_t$. To encode positional information, transformer-based diffusion models [29] often incorporate positional encodings [42] into the self-attention mechanism. However, these encodings are tied to the training sequence length. Consequently, when self-attention is applied to sequences longer than those seen during training, unseen positional embeddings may disrupt spatial understanding and degrade image quality [27].

**Patch-Wise Denoising.** Conventional methods process $Z_t \in \mathbb{R}^{sh \times sw \times d}$ by directly computing self-attention with $Q, K, V \in \mathbb{R}^{sh \times sw \times d}$, which results in a computational cost of $s^4 h^2 w^2 d$ FLOPs. To reduce the significant computational cost and circumvent the resolution limitations introduced by positional encoding, MultiDiffusion [2] divides the input $Z_t$ into $N$ overlapping patches $\{z_t^i\}_{i=1}^N$, where $z_t^i \in \mathbb{R}^{h \times w \times d}$. Then, the denoising network is applied individually to each patch $z_t^i$ to obtain the corresponding denoised patch $z_{t-1}^i$. These individually processed patches are then aggregated to reconstruct the full latent high-resolution $Z_{t-1}$ by averaging the values in the overlapping regions.

Specifically, they apply a shifted crop sampling strategy with strides $S_h$ and $S_w$, corresponding to the height and width dimensions, respectively. As a result, the total number of patches is given by $N = \left(\frac{sh-h}{S_h} + 1\right) \times \left(\frac{sw-w}{S_w} + 1\right)$. This leads to $Q, K, V \in \mathbb{R}^{N \times h \times w \times d}$, resulting in a total computational cost of $Nh^2 w^2 d$ FLOPs in self-attention. In practice, the stride values are typically set to $\left(\frac{h}{2}, \frac{w}{2}\right)$, yielding $N = (2s-1)^2$ patches and a corresponding FLOPs of $(2s-1)^2 h^2 w^2 d$ (Table 1).

### 3.2 Neighborhood Patch Attention

MultiDiffusion circumvents the inherent resolution limitations by decomposing the image into smaller, overlapping patches and processing them individually. They effectively reduce the computational cost in self-attention layers by limiting attention to local regions. However, it often requires substantial overlap between adjacent patches to ensure smooth transitions at the patch boundaries. This overlap causes non-self-attention layers to require nearly 4× more FLOPs under a common stride setting, compared to a single forward pass that processes the full latent at once. A detailed breakdown of these computational costs is provided in Table 1.

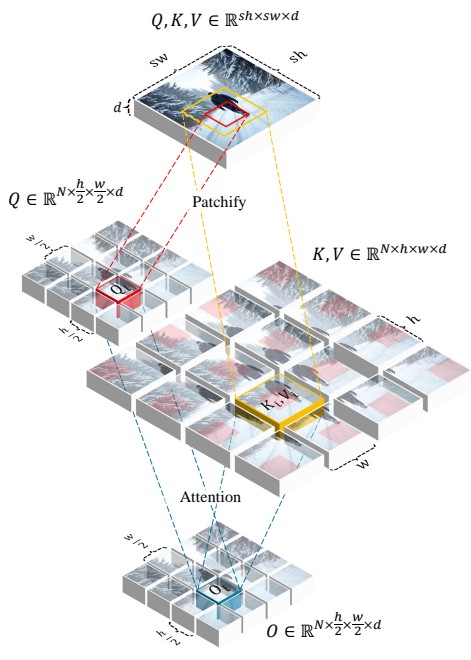

To reduce the computational redundancy, we introduce Neighborhood Patch Attention (NPA) in Figure 2. Our key insight is that layers such as linear, convolution, and cross-attention perform operations on individual tokens or local regions. Unlike self-attention, these layers remain unaffected by increased input resolution, eliminating the need for patch-based processing. Building on this observation, NPA avoids patch-based processing for these non-self-attention layers, allowing them to operate on the full latent tensor $Z_t$ in a single forward pass. This design eliminates redundant computations caused by overlapping patches, thereby keeping the computational cost of non-self-attention layers unchanged (Table 1).

Figure 2: **Process of NPA.**

Within the self-attention mechanism, NPA is designed to further mitigate computational overhead by extracting queries from *non-overlapping* patches. Specifically, given a full query tensor $Q \in \mathbb{R}^{sh \times sw \times d}$, we apply a shifted crop sampling strategy to obtain a set of $N$ query patches $\{Q_i\}_{i=1}^N$, where $Q_i \in \mathbb{R}^{\frac{h}{2} \times \frac{w}{2} \times d}$. The crop stride is set to match the patch size, i.e., $S_h = \frac{h}{2}, S_w = \frac{w}{2}$, resulting in a total of $N = \left(\frac{sh-h/2}{h/2} + 1\right) \times \left(\frac{sw-w/2}{w/2} + 1\right) = 4s^2$ patches. This ensures that the query patches *do not overlap*, keeping the total number of query tokens unchanged. For each non-overlapping query patch $Q_i$, we extract a corresponding key–value patch pair $(K_i, V_i) \in \mathbb{R}^{h \times w \times d}$ from its spatial neighborhood, using a larger window of size $h \times w$ centered on $Q_i$. Because each $K_i$ and $V_i$ patch is

Table 1: **Theoretical FLOPs comparison.** NPA reduces the computational complexity of self-attention without affecting the cost of non–self-attention operations. $k$ denotes the convolution kernel size, and $l$ is the length of text tokens. The Base method represents directly processing the input in a single forward pass. MultiDiffusion is calculated based on a common stride setting. Symbols highlighted in red indicate key elements for comparison.

| Method | Linear | Conv | Cross-Attn | Self-Attn |
|---|---|---|---|---|
| Base [29] | $s^2 hwd^2$ | $s^2 hwk^2 d^2$ | $s^2 hwld$ | $s^4 h^2 w^2 d$ |
| MultiDiffusion [2] | $(2s-1)^2 hwd^2$ | $(2s-1)^2 hwk^2 d^2$ | $(2s-1)^2 hwld$ | $(2s-1)^2 h^2 w^2 d$ |
| NPA(Ours) | $s^2 hwd^2$ | $s^2 hwk^2 d^2$ | $s^2 hwld$ | $s^2 h^2 w^2 d$ |

Figure 3: **Comparison between different reference latents.**

drawn from an expanded spatial window, the overlap between these patches allows every query patch to attend to a wider context and ensures smooth transitions across patch boundaries. We describe the overall process of the Query, Key, and Value patch extraction algorithm in the supplementary materials.

After that, self-attention is computed between non-overlapping query patch $Q_i$ and its corresponding overlapping K/V neighborhood $(K_i, V_i)$, producing the output $O_i \in \mathbb{R}^{\frac{h}{2} \times \frac{w}{2} \times d}$ and resulting in $\frac{h^2 w^2}{4} d$ FLOPs. As this process is computed individually on $4s^2$ patches, the final cost is $s^2 h^2 w^2 d$ (Table 1). Finally, we reassemble the individually computed output patches $\{O_i\}_{i=1}^N$ into the full attention output tensor $O \in \mathbb{R}^{sh \times sw \times d}$ based on their original spatial positions, which is then passed to subsequent layers (e.g., an MLP block).

### 3.3 ScaleDiff Upscaling Pipeline

To maintain the global structure of a low-resolution image while enhancing high-frequency details during image generation, we employ an SDEdit [28]-based pipeline. Starting from a low-resolution image latent $z$, we first upscale it to obtain $Z_{ref}$, then inject noise up to the intermediate time step $\tau$, and apply denoising using NPA. However, naively applying this process often results in outputs that lack fine texture details and appear overly smoothed [18]. This phenomenon arises because upscaling a low-resolution image closely resembles the resizing operation used during training. As a result, the model tends to denoise the input toward the distribution of resized training images, rather than synthesizing fine-grained details [20].

To understand this limitation, we compare two upsampling strategies in Figure 3. Upsampling directly in latent space yields $Z_{LU}$, which lacks high-frequency components, resulting in visible artifacts

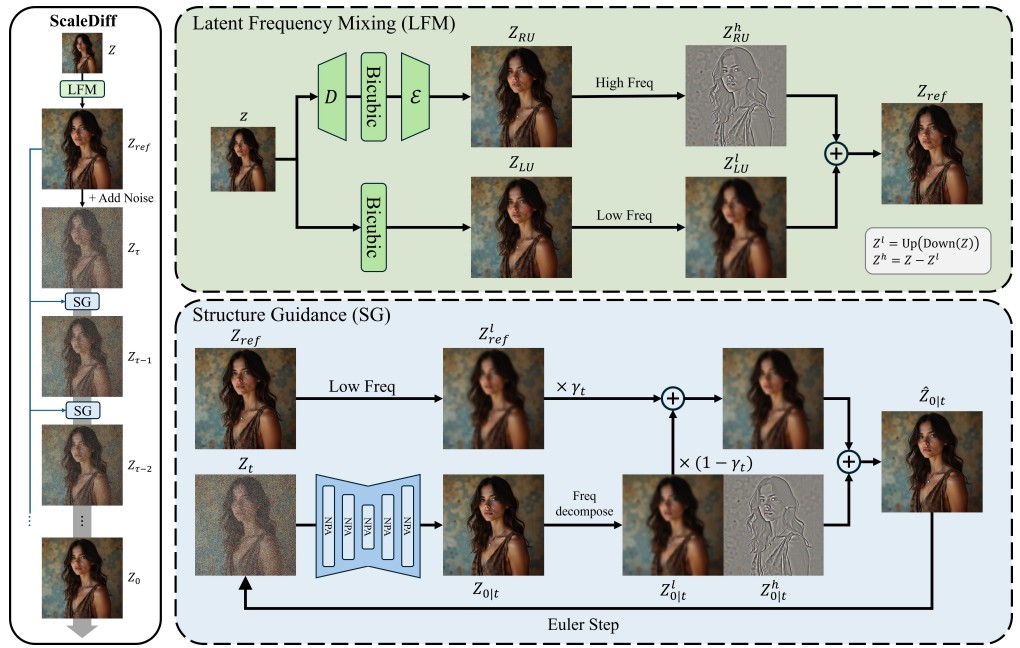

Figure 4: **Overview of our pipeline.** ScaleDiff starts from a generated low-resolution latent, upsamples it with LFM, and diffuses it to an intermediate timestep $\tau$. At each denoising step, the network—integrated with NPA—applies structure guidance to preserve the global image structure.

in the decoded image that propagate to the final output. However, because $Z_{LU}$ deviates from the training distribution of RGB-resized images, the subsequent denoising process is not biased toward oversmoothing. In contrast, upsampling in RGB space followed by VAE encoding yields $Z_{RU}$, which contains rich frequency information and ensures stable, artifact-free decoding. However, since this process closely mimics training-time resizing operations, it strongly biases the model toward reproducing oversmoothed textures instead of generating fine details.

To leverage the complementary strengths of both approaches, we propose **Latent Frequency Mixing (LFM)**. By combining the low-frequency content from $Z_{LU}$—which steers denoising away from the oversmoothing regime—with the high-frequency content from $Z_{RU}$—which ensures stable decoding—we can achieve both sharpness and natural texture. The refined reference latent is:

$$Z_{ref} = Z_{RU}^h + Z_{LU}^l, \tag{3}$$

where $^l$ and $^h$ denote low- and high-frequency components obtained by downsampling and upsampling operations and their residual. This construction guides subsequent denoising toward generating detailed outputs without oversmoothing.

Since NPA processes images through patches, it can introduce repetitive patterns. To mitigate this and enforce global structural consistency, we apply **Structure Guidance (SG)** following prior work [17, 20, 39]. At each timestep $t$, we obtain a clean estimate $Z_{0|t}$ from the noisy latent $Z_t$ and guide it toward $Z_{ref}$ by blending their low-frequency components:

$$\hat{Z}_{0|t} = Z_{0|t}^h + (1 - \gamma_t)Z_{0|t}^l + \gamma_t Z_{ref}^l, \tag{4}$$

where $\gamma_t$ controls the guidance strength. This guided prediction $\hat{Z}_{0|t}$ is then utilized to compute the subsequent noisy latent $Z_{t-1}$. This process steers the generation towards the global structure defined by $Z_{ref}$ while allowing the model to synthesize high-frequency details. Figure 4 illustrates our pipeline.

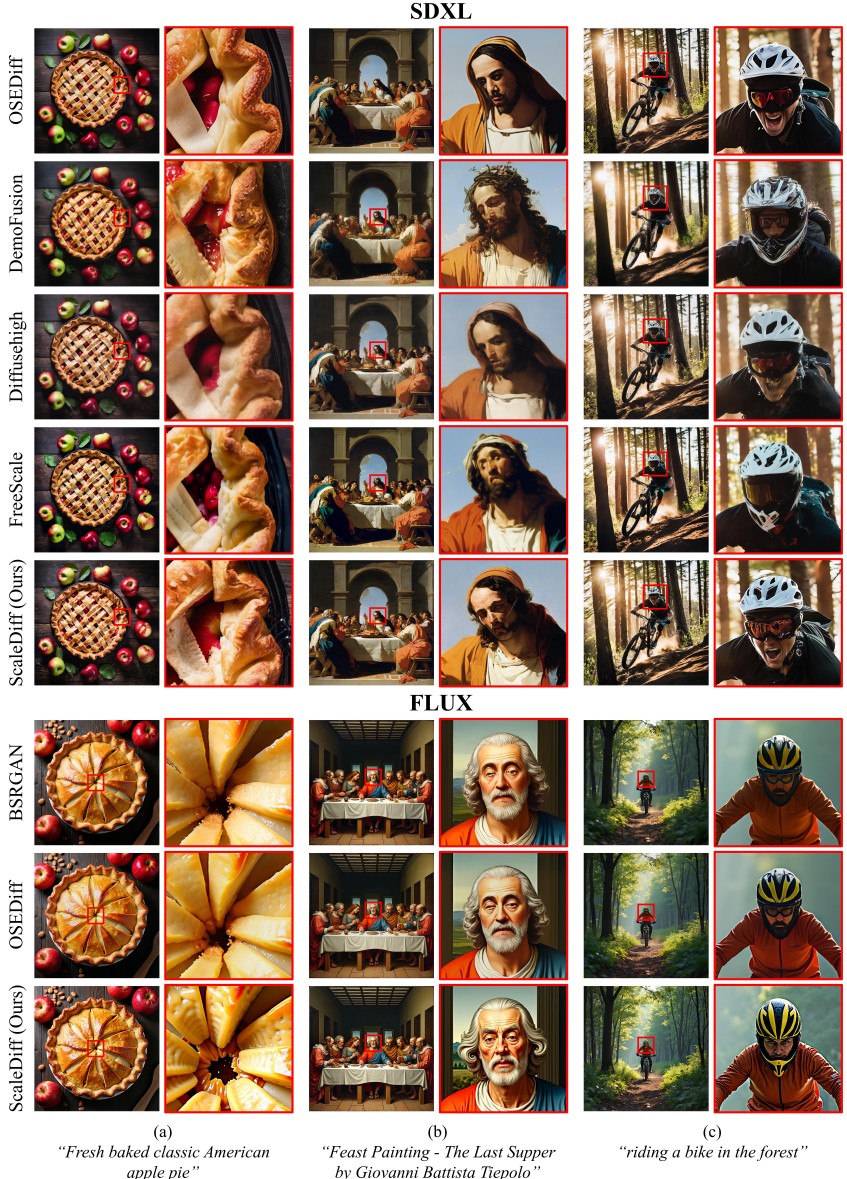

Figure 5: **Qualitative comparison with other methods.** All images are generated at $4096^2$ from the same low-resolution input. Zoom in for a better view.

## 4 Experiments

### 4.1 Experimental Settings

**Implementation Details.** We evaluate our proposed method, **ScaleDiff**, on both FLUX [21] and SDXL [30] within an iterative $1024^2 \rightarrow 2048^2 \rightarrow 4096^2$ generation pipeline. For **FLUX**, we use a noise timestep $\tau = 600$, and a structure guidance strength of $\gamma_t = t$. This setup uses 30 denoising steps with a guidance scale of 3.5. For **SDXL**, we set $\tau = 400$, and the structure guidance strength to $\gamma_t = 1 - \bar{\alpha}_t$. This configuration uses 50 denoising steps with a classifier-free guidance (CFG) [14] scale of 7.5. All experiments are conducted on a single NVIDIA A6000 GPU.

**Baselines.** We compare our method with recent training-free methods (ScaleCrafter [12], HiDiffusion [49], DiffuseHigh [20], FreeScale [31], DemoFusion [8], AccDiffusion v2 [24]), super-resolution models (BSRGAN [48], OSEDiff [46]), and a training-based model UltraPixel [33]. Training-free

Table 2: **Quantitative comparison results**. The best results are shown in **bold**, and the second best results are underlined. All time measurements are expressed in seconds.

| Model | Resolution | Method | FID ↓ | KID ↓ | IS ↑ | FID$_p$ ↓ | KID$_p$ ↓ | IS$_p$ ↑ | CLIP ↑ | Time ↓ |
|---|---|---|---|---|---|---|---|---|---|---|
| SDXL | 2048² | SDXL Direct [30] | 88.56 | 0.0124 | 13.25 | 58.73 | 0.0137 | 20.79 | 31.57 | 47 |
| | | SDXL + BSRGAN [48] | 64.60 | 0.0041 | 18.40 | 41.40 | 0.0092 | 23.19 | 33.03 | **13** |
| | | SDXL + OSEDiff [46] | 64.79 | 0.0046 | 18.89 | 41.76 | 0.0094 | 23.58 | 32.79 | 29 |
| | | UltraPixel [33] | 64.61 | 0.0056 | 18.58 | 42.44 | 0.0093 | 25.15 | 32.61 | 71 |
| | | ScaleCrafter [12] | 68.68 | 0.0033 | 16.56 | 43.46 | 0.0064 | 23.52 | 32.07 | 64 |
| | | HiDiffusion [49] | 69.52 | 0.0040 | 18.22 | 42.92 | 0.0067 | 24.01 | 31.50 | 33 |
| | | DiffuseHigh [20] | 63.27 | 0.0033 | 19.10 | 38.15 | 0.0062 | 24.95 | 32.77 | 45 |
| | | FreeScale [31] | 63.50 | **0.0031** | 19.06 | 38.27 | 0.0062 | 24.25 | 32.62 | 69 |
| | | AccDiffusion v2 [24] | 64.86 | 0.0039 | 18.37 | 38.24 | 0.0068 | 25.66 | 32.62 | 199 |
| | | Demofusion [8] | 63.36 | 0.0032 | 19.15 | **35.98** | **0.0050** | 26.42 | 32.72 | 125 |
| | | ScaleDiff (Ours) | **62.98** | 0.0032 | **19.54** | 38.03 | 0.0067 | 25.70 | **33.11** | 31 |
| | 4096² | SDXL Direct [30] | 182.05 | 0.0717 | 7.99 | 80.80 | 0.0250 | 17.68 | 27.82 | 328 |
| | | SDXL + BSRGAN [48] | 64.88 | 0.0044 | 18.16 | 48.97 | 0.0160 | 17.04 | 33.02 | **14** |
| | | SDXL + OSEDiff [46] | 65.35 | 0.0045 | 18.69 | 45.67 | 0.0118 | 17.61 | 32.88 | 122 |
| | | UltraPixel [33] | 65.39 | 0.0055 | 19.08 | 47.09 | 0.0112 | **20.64** | 32.33 | 386 |
| | | ScaleCrafter [12] | 86.66 | 0.0110 | 15.14 | 79.39 | 0.0217 | 14.47 | 30.25 | 932 |
| | | HiDiffusion [49] | 105.37 | 0.0216 | 13.87 | 112.30 | 0.0494 | 12.22 | 27.21 | 124 |
| | | DiffuseHigh [20] | 63.91 | 0.0034 | 18.99 | 42.30 | 0.0079 | 19.54 | 32.68 | 325 |
| | | FreeScale [31] | 64.33 | 0.0036 | 19.18 | 39.56 | **0.0079** | 18.91 | 32.56 | 517 |
| | | AccDiffusion v2 [24] | 64.64 | 0.0037 | 18.56 | 40.92 | 0.0083 | 18.42 | 32.34 | 1599 |
| | | Demofusion [8] | 65.06 | 0.0041 | 19.13 | 41.29 | 0.0079 | 19.59 | 32.61 | 1005 |
| | | ScaleDiff (Ours) | **61.87** | **0.0025** | 19.56 | **38.89** | 0.0080 | 20.41 | **33.04** | 113 |
| FLUX | 2048² | FLUX Direct [21] | 68.78 | 0.0069 | 18.57 | 42.84 | 0.0086 | 22.46 | 30.79 | 150 |
| | | FLUX + BSRGAN [48] | 64.65 | 0.0052 | **19.07** | 42.01 | 0.0081 | 22.98 | 31.21 | **33** |
| | | FLUX + OSEDiff [46] | 65.10 | 0.0056 | 18.46 | 41.88 | 0.0078 | 23.25 | 31.03 | 46 |
| | | ScaleDiff (Ours) | 64.31 | **0.0047** | 18.51 | **40.03** | 0.0073 | 23.38 | **31.22** | 103 |
| | 4096² | FLUX Direct [21] | 459.07 | 0.2775 | 1.61 | 367.47 | 0.2642 | 1.22 | 18.03 | 1251 |
| | | FLUX + BSRGAN [48] | 64.76 | 0.0051 | 18.84 | 49.30 | 0.0125 | 16.92 | **31.19** | **34** |
| | | FLUX + OSEDiff [46] | 64.22 | 0.0052 | **19.16** | 48.37 | 0.0112 | 16.99 | 31.13 | 136 |
| | | ScaleDiff (Ours) | **64.06** | **0.0044** | 18.36 | **44.29** | **0.0098** | 17.41 | 31.14 | 407 |

baselines are evaluated on SDXL, as they are optimized for U-Net architectures. For FLUX, we compare against the base model (natively supports resolutions up to 2048²) and SR methods.

**Evaluation.** For quantitative evaluation, we randomly sample 1,000 image-text pairs from the LAION-5B [38] dataset and generate one image per prompt using each method. We compute the Fréchet Inception Distance (FID) [13], Kernel Inception Distance (KID) [4], and Inception Score (IS) [37] between generated images and real images. However, these metrics typically require resizing images to 299² pixels, thereby limiting the evaluation of fine-grained details. To better assess detail fidelity, we extract multiple patches from each image and calculate patch-level FID$_p$, KID$_p$, and IS$_p$ following [8]. We also measure the CLIP Score [32] to evaluate text-image alignment.

## 4.2 Quantitative Comparison

Table 2 compares ScaleDiff with baseline methods for generating images at 2048² and 4096² resolutions. On SDXL, ScaleDiff consistently outperforms existing training-free, training-based, and super-resolution methods across key quality metrics, demonstrating its ability to generate high-fidelity images. Similar results on FLUX further confirm ScaleDiff's robustness and model-agnostic design.

ScaleDiff also achieves remarkable inference efficiency. For 4096² resolution on SDXL, it requires only 113 seconds—the fastest among training-free methods. Compared to the patch-based method Demofusion, ScaleDiff achieves an 8.9× speedup while surpassing it in most evaluation metrics, demonstrating NPA's effectiveness. On FLUX, applying NPA yields a 3.1× speedup over direct inference at 4096² resolution. While super-resolution models like BSRGAN offer faster inference, they struggle to produce fine details, as reflected in lower patch-level scores. In contrast, ScaleDiff successfully balances generative quality with computational efficiency.

## 4.3 Qualitative Comparison

Figure 5 presents a qualitative comparison of ScaleDiff with baseline methods for 4096² image generation. While all methods produce high-quality outputs, prior approaches exhibit notable limitations. Super-resolution models like BSRGAN and OSEDiff fail to reproduce fine details, resulting in visibly corrupted facial features (Figure 5b,c). DemoFusion effectively generates fine

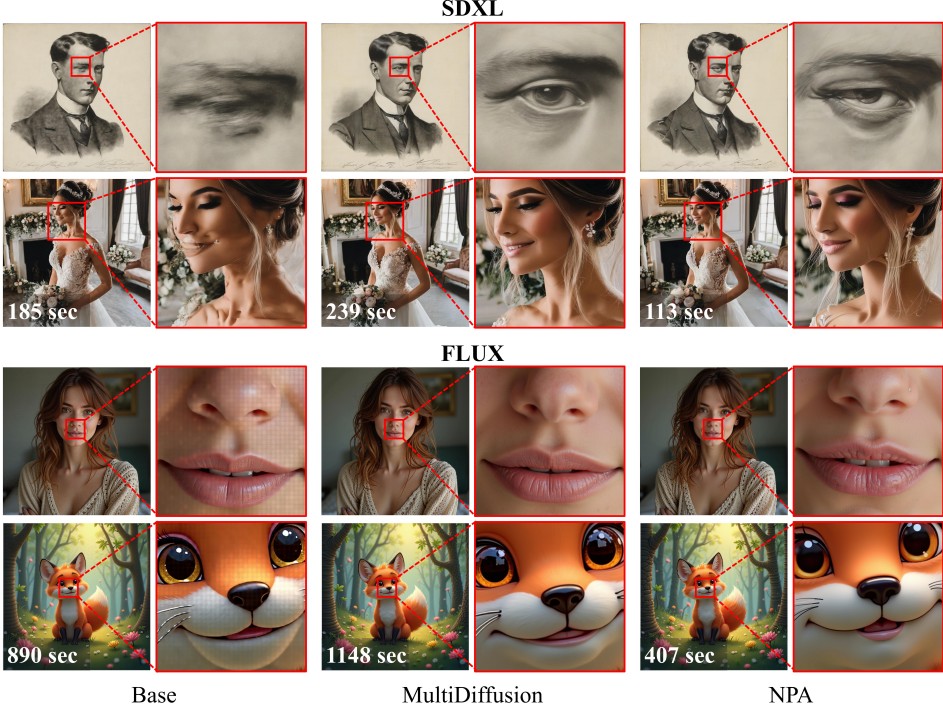

Figure 6: **Qualitative comparison of replacing NPA in the ScaleDiff pipeline.** Inference time for each method is shown in the bottom left. All images are generated at $4096^2$ resolution.

Table 3: **Quantitative results of ablation study**.

| Attention | $LFM$ | $SG$ | FID $\downarrow$ | KID $\downarrow$ | IS $\uparrow$ | $FID_p \downarrow$ | $KID_p \downarrow$ | $IS_p \uparrow$ | CLIP $\uparrow$ | Time $\downarrow$ |
|---|---|---|---|---|---|---|---|---|---|---|
| Base | ✓ | ✓ | 61.91 | 0.0028 | 19.47 | 39.94 | 0.0082 | 20.09 | 33.01 | 185 |
| MultiDiffusion | ✓ | ✓ | **61.71** | **0.0021** | **19.71** | **38.08** | **0.0069** | **20.85** | 33.04 | 239 |
| NPA | ✓ | ✓ | 61.87 | 0.0025 | 19.56 | 38.89 | 0.0080 | 20.41 | **33.04** | **113** |
| NPA | | | 64.17 | 0.0036 | 19.49 | 41.55 | 0.0092 | 19.41 | 33.02 | 113 |
| NPA | ✓ | | 62.34 | 0.0028 | 19.19 | 39.49 | 0.0085 | 20.16 | 33.01 | 113 |
| NPA | | ✓ | 64.12 | 0.0035 | 18.86 | 41.50 | 0.0091 | 19.71 | 33.04 | 113 |
| NPA | ✓ | ✓ | **61.87** | **0.0025** | **19.56** | **38.89** | **0.0080** | **20.41** | **33.04** | 113 |

details but often suffers from repetitive object patterns due to its patch-based processing (Figure 5c). DiffuseHigh lacks detailed textures due to inherent constraints of RGB-space upsampling (Figure 5a), In contrast, ScaleDiff produces results with improved global structure and finer details, demonstrating superior qualitative performance across different models.

## 4.4 Ablation Study

**Effectiveness of NPA.** We validate the effectiveness of our proposed Neighborhood Patch Attention (NPA) by comparing it against two alternatives integrated into the ScaleDiff pipeline: (1) direct high-resolution inference (Base) and (2) a standard patch-based method (MultiDiffusion). As shown in Figure 6 and Table 3, all methods maintain a stable global structure, likely due to the shared ScaleDiff pipeline. However, the Base method produces local artifacts on SDXL and lacks fine details on FLUX, while requiring significantly longer inference times. MultiDiffusion generates high-quality images and achieves the best scores, but suffers from substantial computational overhead (1148s on FLUX). In contrast, our NPA achieves scores comparable to MultiDiffusion while being more efficient (407s on FLUX, $2.8\times$ speedup), demonstrating an effective balance between generation quality and computational efficiency.

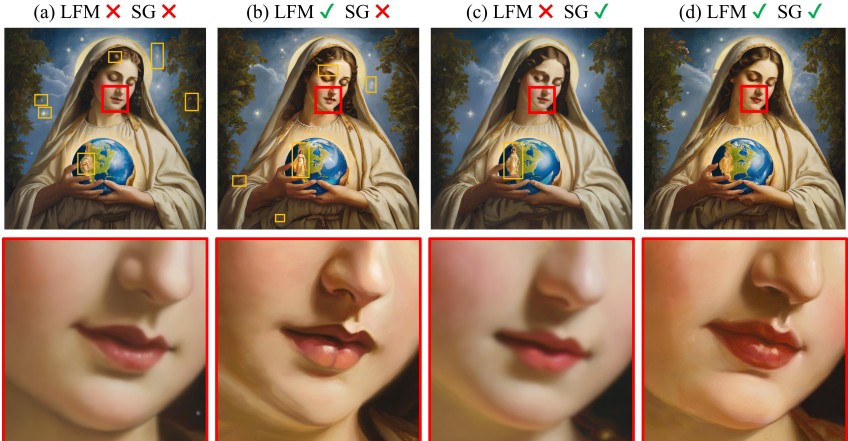

(a) LFM ✗ SG ✗      (b) LFM ✓ SG ✗      (c) LFM ✗ SG ✓      (d) LFM ✓ SG ✓

Figure 7: **Ablating each component of ScaleDiff. The yellow box highlights the repetition artifacts. All images are generated at** $4096^2$ **using SDXL [30]. Zoom in for a better view.**

Table 4: **Ablating noise timestep** $\tau$.

| Model | $\tau$ | FID↓ | KID↓ | IS↑ | $\text{FID}_p$↓ | $\text{KID}_p$↓ | $\text{IS}_p$↑ | CLIP↑ |
|---|---|---|---|---|---|---|---|---|
| SDXL | 700 | 63.61 | 0.0036 | 18.93 | **37.65** | **0.0064** | 25.02 | 33.01 |
| | 600 | 63.60 | 0.0034 | 19.20 | 37.94 | 0.0067 | **25.15** | 33.06 |
| | 500 | 63.30 | 0.0033 | 19.43 | 38.77 | 0.0073 | 24.42 | **33.09** |
| | 400 | **63.12** | **0.0032** | **19.36** | 38.33 | 0.0070 | 24.56 | 33.07 |
| | 300 | 63.22 | 0.0032 | 19.56 | 39.38 | 0.0077 | 23.85 | 33.07 |
| FLUX | 700 | 64.44 | 0.0049 | 18.22 | **40.37** | **0.0073** | **23.62** | 31.27 |
| | 600 | 64.45 | 0.0049 | **18.52** | 40.76 | 0.0076 | 23.16 | 31.29 |
| | 500 | 64.13 | 0.0049 | 18.36 | 41.72 | 0.0081 | 22.78 | 31.29 |
| | 400 | **63.59** | **0.0047** | 18.45 | 41.99 | 0.0083 | 22.90 | 31.25 |
| | 300 | 63.67 | 0.0048 | 18.30 | 42.07 | 0.0084 | 23.24 | **31.33** |

**Effectiveness of LFM and SG.** In Figure 7 and Table 3, we validate the contributions of Latent Frequency Mixing (LFM) and Structure Guidance (SG). When both components are removed (Fig. 7a), the model fails to generate coherent results, exhibiting severe object repetition and heavily oversmoothed textures. Adding LFM alone (Fig. 7b) reduces oversmoothing and enables the synthesis of finer details, which is confirmed by improvements in patch-level metrics in Table 3. Applying SG alone (Fig. 7c) effectively mitigates object repetition, demonstrating its role in enforcing global structural coherence. The full ScaleDiff pipeline (Fig. 7d), which combines both LFM and SG, concurrently addresses both issues and achieves the best overall performance among all ablated configurations.

**Ablation of Noise Timestep** $\tau$**.** The noise timestep $\tau$ is a critical hyperparameter governing the trade-off between preserving global structure from the upsampled reference image (lower $\tau$) and enabling sufficient synthesis of fine-grained details (higher $\tau$). We conduct an ablation study to determine the optimal $\tau$ for both architectures. As detailed in Table 4, $\tau = 400$ for SDXL and $\tau = 600$ for FLUX provide the best balance between structural fidelity and detail generation.

## 5 Conclusion

In this paper, we propose ScaleDiff, an efficient and model-agnostic framework that enhances the resolution capabilities of pretrained diffusion models without requiring additional training. We introduce Neighborhood Patch Attention (NPA), a mechanism that significantly reduces the computational redundancy typical of traditional patch-based diffusion approaches. In addition, we propose Latent Frequency Mixing (LFM) and incorporate Structure Guidance (SG) within an upsample–diffuse–denoise pipeline to improve fine detail synthesis and structural consistency. Our experiments, conducted on both U-Net and Diffusion Transformer architectures, show that ScaleDiff achieves state-of-the-art performance among training-free methods, delivering superior image quality and faster inference across diverse models. These results highlight ScaleDiff as a powerful and versatile solution for higher-resolution image generation.

**Acknowledgment**. This was partly supported by the Institute of Information & Communications Technology Planning & Evaluation (IITP) grant funded by the Korean government(MSIT) (No.RS-2020-II201373, Artificial Intelligence Graduate School Program(Hanyang University)) and the Institute of Information &Communications Technology Planning & Evaluation (IITP) grant funded by the Korean government(MSIT) (No.RS-2025-02215122, Development and Demonstration of Lightweight AI Model for Smart Homes).

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

# A  Additional Implementation Details

When generating images with various aspect ratios, we ensure that the longer side of the initial image matches the model's trained resolution. For frequency decomposition, we use a spatial downsampling ratio of 4 for FLUX and 8 for SDXL. When evaluating the inference speed of MultiDiffusion [2], the overlap ratio is set to 50%, and we use a batch size of 16 for SDXL and 1 for FLUX.

In U-Net-based models such as SDXL [30], the spatial resolution is progressively downsampled across layers. Accordingly, we also downsample the native resolution $(h, w)$ of NPA to match the downsampling ratio at each corresponding layer. In FLUX [21], the MM-DiT architecture [9] is used, where the text tokens are concatenated with latent tokens and jointly processed through a self-attention layer. Accordingly, in NPA, the text tokens are duplicated for each patch and concatenated with the corresponding latent tokens within each patch. After the NPA processing, the text tokens are averaged across all patches. In the original setting, text tokens are assigned the position $(0, 0)$ in RoPE [42]. When duplicating the text tokens, we assign them the position of the top-left corner of the corresponding Key/Value patch to ensure proper spatial processing.

# B  Additional Details and Experiments on Neighborhood Patch Attention

## B.1  Query Window Random Shifting

While Neighborhood Patch Attention utilizes overlapping key/value patches to ensure a smooth transition at patch boundaries, minor boundary artifacts can sometimes appear in the generated output. **Query Window Random Shifting** is an optional technique designed to further alleviate such artifacts by introducing random variations to the query patch grid at each layer (Figure 8). Specifically, we randomly sample the top and left offsets for padding uniformly from the respective ranges $[0, \frac{h}{2}]$ and $[0, \frac{w}{2}]$. The query tensor is then zero-padded by a total of $\frac{h}{2}$ in height and $\frac{w}{2}$ in width using these randomly sampled top-left offsets. Query patches are subsequently extracted from this enlarged canvas. After attention computation, regions corresponding to the added padding are discarded. Since offsets are independently resampled at each layer, explicit patch boundaries are avoided, which reduces border artifacts with minimal computational overhead. Note that this technique was not used during the evaluation in this paper.

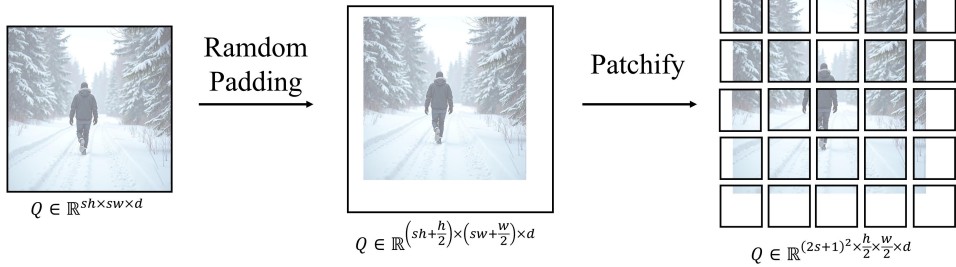

Figure 8: **Illustration of Query Window Random Shifting.**

## B.2  Comparison of Generation time

Figure 9 presents the model processing time at various resolutions for SDXL and FLUX, comparing three methods: Direct Inference (Base), MultiDiffusion, and NPA. Direct Inference shows a quadratic growth in processing time as resolution increases, primarily due to the cost of global self-attention. MultiDiffusion achieves linear scaling with resolution but suffers from higher baseline overhead, caused by redundant computation on overlapping patches. In contrast, NPA eliminates such redundancy and maintains linear scaling, resulting in the lowest processing time.

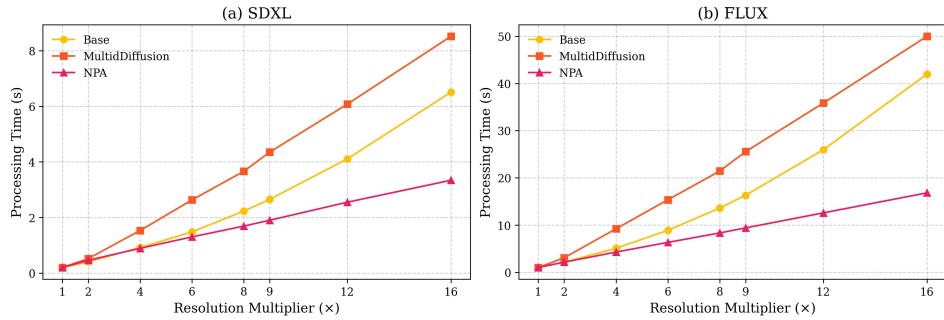

Figure 9: **Comparison of model processing time.** A resolution multiplier of $1\times$ corresponds to generation at $1024^2$ resolution.

## B.3 Panorama Generation with NPA

Our NPA adopts the behavior of MultiDiffusion, making it suitable for a wide range of applications. Notably, Figure 10 presents the results of using NPA for panorama generation on FLUX. No other ScaleDiff components were used.

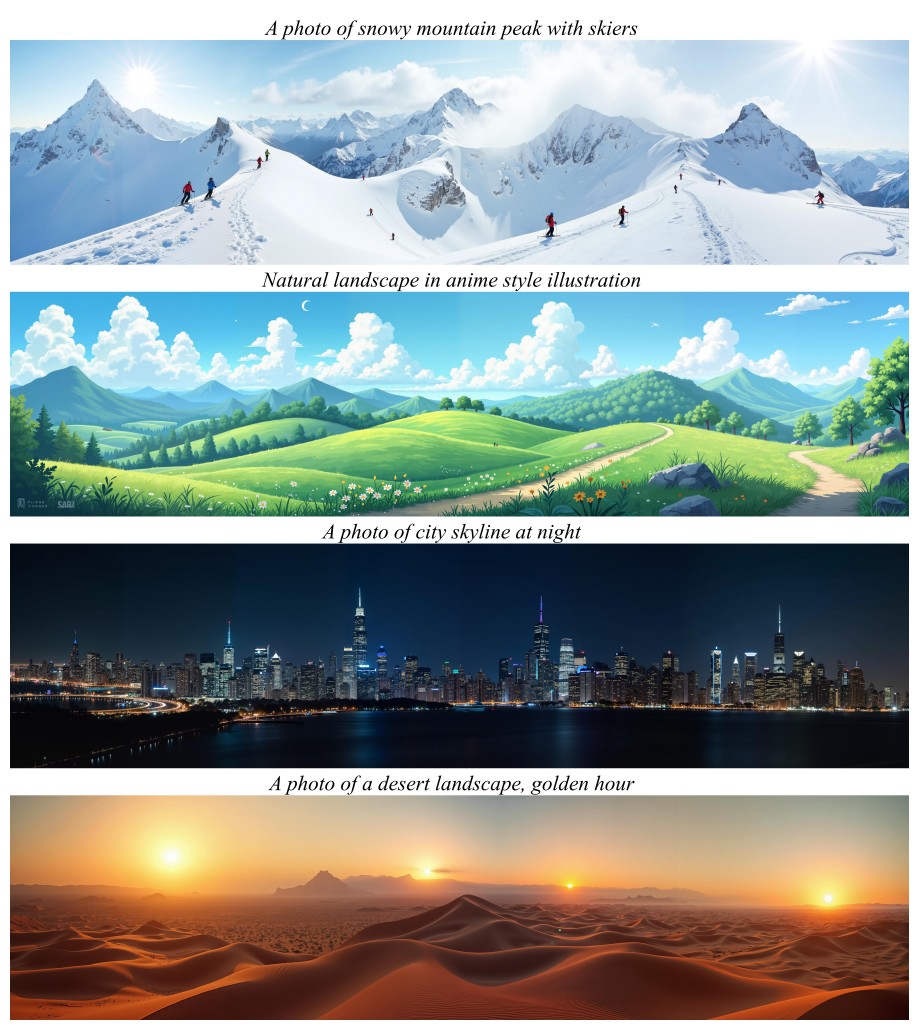

Figure 10: **Panorama Generation with NPA.** All images are generated at $1024 \times 4096$ resolution.

## B.4 Patch Extraction Algorithm of NPA

Algorithm 1 presents the detailed patch extraction procedure for NPA. Note that query window random shifting (Section B.1) is not included in this algorithm.

---

**Algorithm 1** NPA: Query/Key/Value Patch Extraction

---

1: **Input:** $\mathbf{Q}, \mathbf{K}, \mathbf{V} \in \mathbb{R}^{mh \times nw \times d}$         ▷ Full query, key, value tensor
2: **Parameters:** $h, w$         ▷ Native height and width
3: **Output:** $\{\mathbf{Q}_i\}_{i=1}^{N}$         ▷ Set of non-overlapping query patches
4:         $\{\mathbf{K}_i\}_{i=1}^{N}, \{\mathbf{V}_i\}_{i=1}^{N}$         ▷ Set of overlapping key, value patches

5: $N_r \leftarrow \frac{mh - h/2}{h/2} + 1$         ▷ Number of patch rows
6: $N_c \leftarrow \frac{nw - w/2}{w/2} + 1$         ▷ Number of patch columns
7: $N \leftarrow N_r \times N_c$         ▷ Total number of patches
8: **for** $i \leftarrow 1$ to $N$ **do**
9:     $h_{start}^{q} \leftarrow \lfloor i/N_r \rfloor \times \frac{h}{2}$         ▷ Top-left coordinate of the query patch
10:     $w_{start}^{q} \leftarrow (i \mod N_r) \times \frac{w}{2}$
11:     $h_{end}^{q} \leftarrow h_{start}^{q} + \frac{h}{2}$         ▷ Bottom-right coordinate of the query patch
12:     $w_{end}^{q} \leftarrow w_{start}^{q} + \frac{w}{2}$
13:     $h_{start}^{kv} \leftarrow \text{clamp}(h_{start}^{q} - \frac{h}{4}, 0, sh - h)$         ▷ Center K/V patch around query patch
14:     $w_{start}^{kv} \leftarrow \text{clamp}(w_{start}^{q} - \frac{w}{4}, 0, sw - w)$         ▷ Clamp for window shifting at the edge
15:     $h_{end}^{kv} \leftarrow h_{start}^{kv} + h$
16:     $w_{end}^{kv} \leftarrow w_{start}^{kv} + w$
17:     $\mathbf{Q}_i \leftarrow \mathbf{Q}[h_{start}^{q} : h_{end}^{q}, w_{start}^{q} : w_{end}^{q}, :]$         ▷ Non-overlapping query patch extraction
18:     $\mathbf{K}_i \leftarrow \mathbf{K}[h_{start}^{kv} : h_{end}^{kv}, w_{start}^{kv} : w_{end}^{kv}, :]$         ▷ Overlapping key, value patch extraction
19:     $\mathbf{V}_i \leftarrow \mathbf{V}[h_{start}^{kv} : h_{end}^{kv}, w_{start}^{kv} : w_{end}^{kv}, :]$
20: **end for**
21: **return** $\{\mathbf{Q}_i\}_{i=1}^{N}, \{\mathbf{K}_i\}_{i=1}^{N}, \{\mathbf{V}_i\}_{i=1}^{N}$

---

# C Qualitative Results on Various Models

We present qualitative results of ScaleDiff using SDXL and FLUX across various aspect ratios and resolutions in Figure 11 and Figure 12. To highlight the model-agnostic nature of our method, we also include results using Lumina-T2X [10] in Figure 13.

# D Limitation

ScaleDiff has some limitations. First, as a tuning-free framework, its performance is inherently constrained by the capabilities of the underlying diffusion model. Second, being a patch-based approach, it relies heavily on the diffusion model's prior knowledge of cropped image regions. This can sometimes lead to inconsistent local content when generating sharp close-up images. Finally, repetitive artifacts may still occur in background regions, a common drawback of patch-based generation methods.

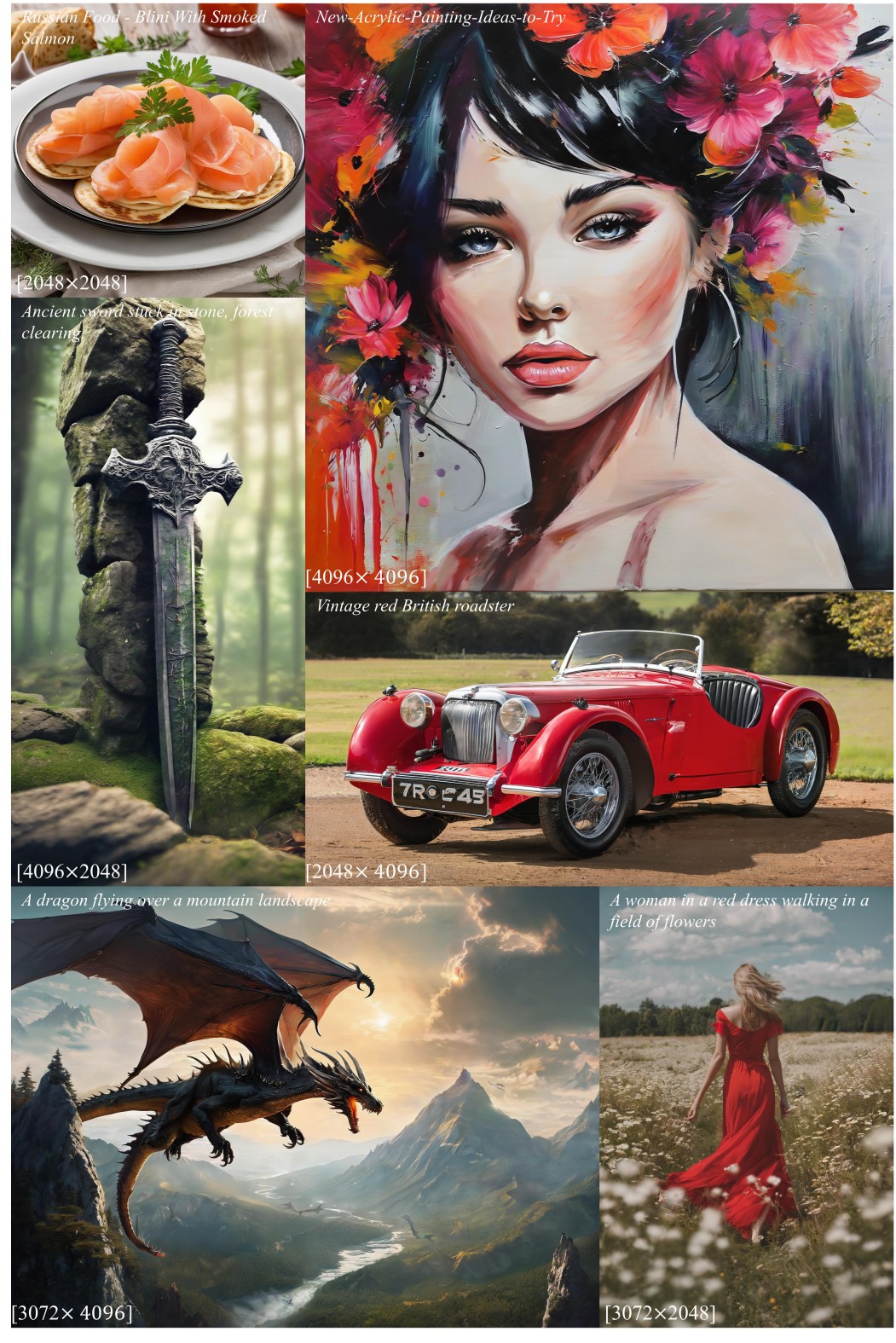

Figure 11: **Qualitative results of ScaleDiff on SDXL**

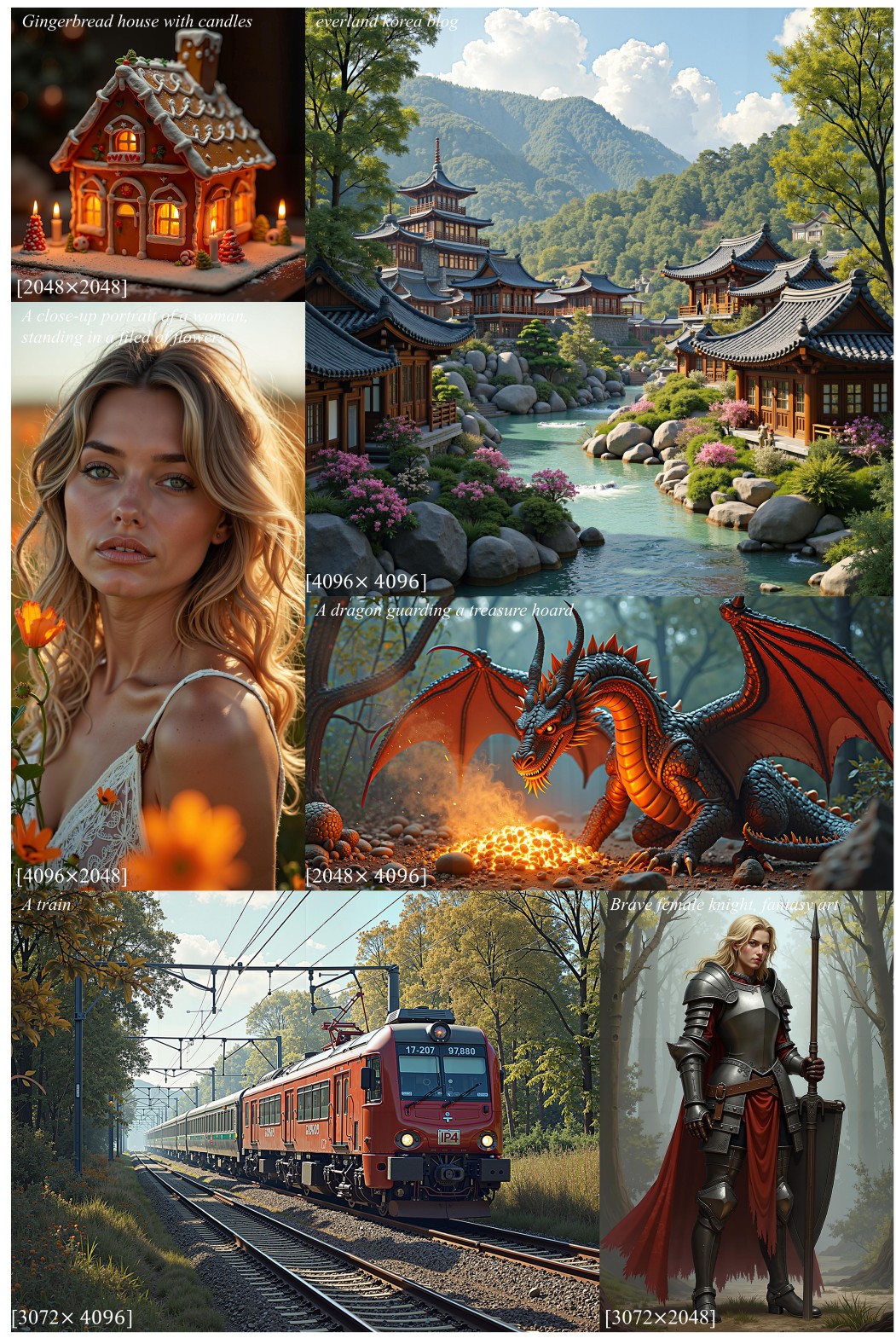

Figure 12: **Qualitative results of ScaleDiff on FLUX**

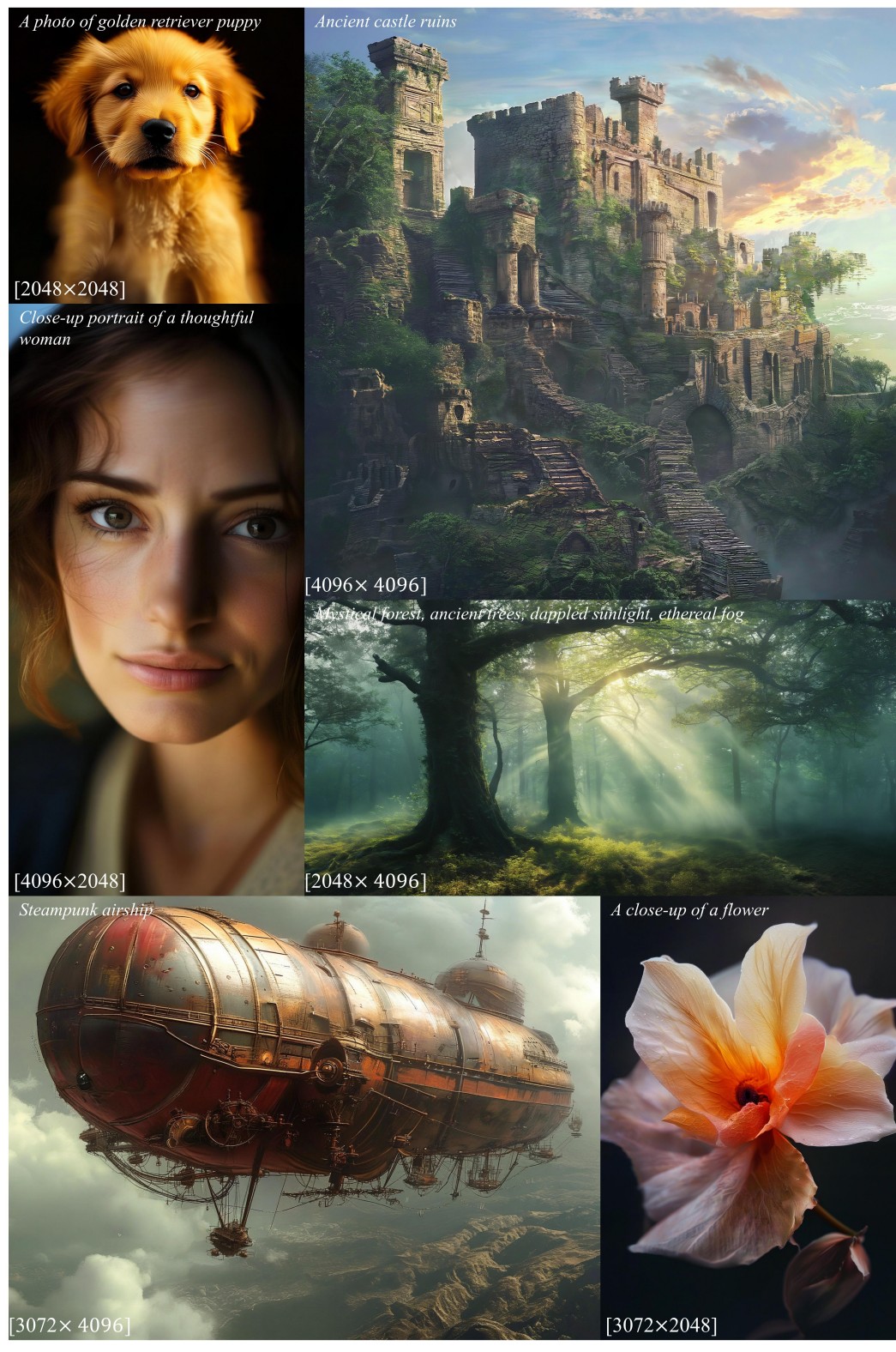

Figure 13: **Qualitative results of ScaleDiff on Lumina-T2X**

