# OpenReview forum: "ScaleDiff: Higher-Resolution Image Synthesis via Efficient and Model-Agnostic Diffusion"
_NeurIPS.cc/2025/Conference — NeurIPS 2025 poster_

### Official Review · Reviewer_BUQE · 2025-06-20

**Clarity:** 2
**Significance:** 2
**Originality:** 2
**Rating:** 3
**Confidence:** 4

**Summary:**

This paper introduces **ScaleDiff**, a training-free, model-agnostic framework for extending the resolution of pretrained diffusion models beyond their original training size.

The key component, **Neighborhood Patch Attention (NPA)**, efficiently reduces redundancy in self-attention by using non-overlapping patches, making it suitable for both U-Net and Diffusion Transformer architectures.

 Combined with **Latent Frequency Mixing (LFM)** for fine-detail generation and **Structure Guidance** for improving global structure, ScaleDiff achieves state-of-the-art image quality and inference speed among training-free upsampling methods.

**Questions:**

### Questions and Suggestions for the Authors

**Q1: Clarify and support the claims on inference speed and performance.**
- Table 2 shows only marginal performance gains or even worse results compared to FLUX+GAN, and the inference speed is notably slower.
- Could the authors provide a more detailed and transparent comparison, including runtime benchmarks under consistent conditions?
- Please clarify or temper the claims regarding state-of-the-art speed and performance if the evidence does not fully support them.

**Q2: Explain the notation and role of `Z_rgb` clearly.**
- The notation `Z_rgb` is ambiguous—is it a latent representation or raw RGB pixel data?
- Given that other `Z_` variables denote latent features, please clarify this inconsistency to avoid reader confusion.
- Consider adding a dedicated explanation or diagram showing how `Z_rgb` differs from or relates to `Z_latent`.

**Q3: Justify the construction of the reference latent `Z_ref` as the sum of `Z_high` and `Z_low`.**
- What is the motivation behind this design choice?
- Would using the actual RGB image as the reference latent provide a clearer or stronger supervisory signal?
- Please discuss alternatives and provide ablation studies or analyses demonstrating why this approach is preferable.

**Q4: Explain the large difference in timestep parameter `tau` between backbones.**
- Why is `tau=0.6` used for FLUX while `tau=400` is used for SDXL?
- Since timesteps are typically integers, please clarify what these values represent and why such a large gap is necessary.
- Providing more details on how `tau` is chosen and tuned for different backbones would improve reproducibility.

**Q5: Improve qualitative results and visual assessments.**
- The generated images in Figure 4 appear random, making it difficult to visually assess improvements.
- Could the authors provide additional visual comparisons with controlled conditions or include quantitative perceptual metrics to better demonstrate quality differences?

**Ethical Concerns:**

["NO or VERY MINOR ethics concerns only"]

**Final Justification:**

The rebuttal addressed some of my concerns, but not fully satisfactorily. While Lines 53–55 mention the focus on "training-free" methods, Table 2 does not clearly indicate which of the compared methods fall under this category. It remains unclear whether some of the bolded values correspond to training-free approaches or pretrained models, for instance, are methods like DemoFusion and UltraPixel training-free?

Regarding qualitative results, I expected to see more controlled comparisons, for example, using the same random seed and input conditions, to better evaluate whether the proposed method provides a visual improvement that aligns with the quantitative metrics. This is especially important when comparing with methods that perform upscaling from low-resolution inputs.

Overall, based on these unresolved issues and after reading the other reviews, I am inclined to maintain my original score.

**Limitations:**

No, they didn't discuss.

**Quality:**

2

**Strengths And Weaknesses:**

**Strengths**
- The paper is complete and generally clear for a broad readership.
- The method does not require costly retraining, making it more practical.
- Experimental results indicate that the approach is promising.

**Weaknesses**
- The claim of state-of-the-art inference speed and performance is not fully supported. In Table 2, performance improvements over competing methods are marginal or absent, and in some cases, ScaleDiff performs worse than FLUX+GAN. Additionally, the reported inference speed is significantly slower than FLUX+GAN, which contradicts the stated contributions.
- The notation for `Z_rgb` is confusing. It is unclear whether this denotes a latent representation or pixel-level RGB data. Since the main text consistently treats `Z_` variables as latents, this inconsistency reduces clarity.
- The motivation behind constructing `Z_ref` by adding `Z_high` and `Z_low` is unclear. Wouldn’t using the actual RGB image as a reference latent provide a clearer supervisory signal than this manipulated combination?
- There is a large discrepancy in the `tau` parameter between the FLUX backbone (`tau=0.6`) and SDXL (`tau=400`). Since timestep values are typically integers, the reason for such a wide gap should be explained.
- The generated images in Figure 4 appear random, making it difficult to visually assess the superiority of the proposed method.
- Minor issue: there is a typo “SclaeDiff” in line 92.

---

> ### Author Rebuttal · Authors · 2025-07-31
>
> Thank you for your detailed feedback and for raising several important questions. We appreciate that you recognized our method's practicality in avoiding costly retraining. We have carefully addressed your concerns below, focusing on clarifying our core claims and experimental methodology. We hope these explanations provide the necessary clarity and fully resolve your concerns.
>
> >**Q1: Clarify and support the claims on inference speed and performance.**
>
>  Please note that BSRGAN is a “trained” super-resolution model. As stated in our introduction (Lines 53-55), we would like to clarify that our model is the SOTA and the fastest among the “training-free” high-resolution generation methods. Also, while BSRGAN is faster, our model significantly outperformed BSRGAN both qualitatively (Figure 4a) and quantitatively (44.29 vs. 49.30 in patch-level FID score in Table 2).
>
> >**Q2: Explain the notation and role of Z\_rgb clearly.**
>
> In this paper, we denote Z (both Z\_rgb and Z\_latent) as a latent representation. As illustrated in Figure 3, Z\_rgb is a latent representation obtained by first decoding a low-resolution latent into the RGB space, upsampling it in the RGB space, and then re-encoding it back into the latent space. Z\_latent is the result of directly upsampling the low-resolution latent in the latent space. We will revise section 3.3  in the final version upon acceptance to ensure clarity and prevent any misunderstanding.
>
> >**Q3: Justify the construction of the reference latent Z\_ref as the sum of Z\_high and Z\_low.**
>
> The motivation and a detailed analysis of our Latent Frequency Mixing (LFM) technique are provided in the supplementary material (Appendix C). Our analysis shows that using Z\_rgb as the starting point for the diffuse-denoise process leads to an oversmoothing bias in the final image. Conversely, Z\_latent successfully avoids the oversmoothing bias. However, it lacks sufficient high-frequency components, which causes noticeable artifacts whend decoded.
> LFM leverages the low-frequency components from Z\_latent as a base to avoid oversmoothing, while simultaneously replacing its deficient high-frequency components with those from Z\_rgb to resolve the artifact issue.
>
> Our paper's ablation study shows that applying LFM mitigates oversmoothing compared to using RGB image as the reference (Z\_rgb). This is demonstrated visually in Figure 5 (a) vs. (b) and supported quantitatively by significantly better scores on detail metrics in Table 3 (a) vs. (b).
>
> To provide further quantitative evidence of LFM, we conducted an additional experiment comparing the reference latents. We measure PSNR for fidelity and Image Entropy and Mean Variance of Laplacian (mVoL) to evaluate sharpness and the degree of oversmoothing. For the full experimental setup, please refer to our response to Reviewer 2Ebs (Q4).
>
> |  | PSNR↑ | Entropy↑ | mVoL↑ |
> | :---- | :---- | :---- | :---- |
> | Z\_latent | 20.14 | 7.12 | **525.78** |
> | Z\_rgb | **23.58** | 7.07 | 49.69 |
> | **LFM (Ours)** | 21.42 | **7.16** | 382.67 |
>
> The table quantitatively confirms the oversmoothing issue with the Z\_rgb reference (indicated by its low Entropy and mVoL scores) and Z\_latent avoids this issue. Our LFM successfully combines the strengths of both, achieving favorable scores across all metrics.
>
> >**Q4: Explain the large difference in timestep parameter tau between backbones.**
>
>  SDXL is a traditional diffusion model that uses discrete integer timesteps (typically from 1 to 1000), while FLUX  is a Rectified Flow model that operates on a continuous normalized timestep from 0 to 1\. Our notation for FLUX (τ=0.6) represents the starting point of the denoising process as 60% along the continuous-time trajectory. For SDXL, τ=400 means starting from the 400th discrete step. We will clarify the notation in the final version upon acceptance.
> The choice of timestep τ is a critical hyperparameter that balances two competing factors: preserving the global structure of the upsampled image versus allowing the model sufficient freedom to synthesize fine-grained details. The values we selected were determined empirically to find an optimal trade-off point. To thoroughly validate this choice and enhance reproducibility, we will provide a full ablation study in the final version—including both qualitative and quantitative results—to demonstrate the impact of varying τ on generation quality.
>
> >**Q5: Improve qualitative results and visual assessments.**
>
> When comparing against methods that generate images directly at high resolution (e.g., ScaleCrafter), using the same seed does not guarantee similar outputs. This technical challenge is also reflected in the figures of prior works \[1, 2, 3\]. However, for methods that are based on upscaling a low-resolution image (e.g., DiffuseHigh, DemoFusion, and ours), a direct comparison from the same starting image is possible. We will add a new figure with such a controlled comparison in the final version upon acceptance to better highlight the qualitative advantages of our method.
>
> >**Minor Issue: Typo "SclaeDiff" in line 92\.**
>
> Thank you for catching this. We will correct it and also carefully check for any other typos throughout the paper.
>
> \[1\] DemoFusion: Democratising High-Resolution Image Generation With No $$$, in CVPR 2024
> \[2\] Is One GPU Enough? Pushing Image Generation at Higher-Resolutions with Foundation Models, in NeurIPS 2024
> \[3\] FreCaS: Efficient Higher-Resolution Image Generation via Frequency-aware Cascaded Sampling, in ICLR 2025

---

> > ### Comment · Reviewer_BUQE · 2025-08-05
> >
> > Thanks authors for their efforts. The rebuttal addressed some of my concerns, but not fully satisfactorily. While Lines 53–55 mention the focus on "training-free" methods, Table 2 does not clearly indicate which of the compared methods fall under this category. It remains unclear whether some of the bolded values correspond to training-free approaches or pretrained models, for instance, are methods like DemoFusion and UltraPixel training-free?
> >
> > Regarding qualitative results, I expected to see more controlled comparisons, for example, using the same random seed and input conditions, to better evaluate whether the proposed method provides a visual improvement that aligns with the quantitative metrics. This is especially important when comparing with methods that perform upscaling from low-resolution inputs.
> >
> > Overall, based on these unresolved issues and after reading the other reviews, I am inclined to maintain my original score.

---

> > > ### Author Response · Authors · 2025-08-06
> > >
> > > Thank you for your follow-up comment to further improve our manuscript.
> > >
> > > In Table 2, DemoFusion and UltraPixel are "training-free" and "training-based", respectively (Lines 221-223). We will clarify this in the table as well in the final version upon acceptance.
> > >
> > > Also, as promised in our rebuttal, we will add a new figure for a controlled comparison. This figure will use the same starting image as DiffuseHigh and DemoFusion to further highlight our method's superior performance in generating fine details.
> > >
> > > Please let us know if there is anything we can further clarify or discuss for the positive reassessment of our paper.

---

> ### Author Response · Authors · 2025-08-05
>
> Dear Reviewer BUQE,
>
> Thank you once again for your thoughtful and constructive review. As the discussion deadline approaches, we wanted to kindly check if you had any additional questions or concerns we could clarify. We truly appreciate your time and effort during this busy period and would be happy to provide further details on any aspect of our work.
>
> Best regards,
> Authors

---

### Official Review · Reviewer_YvLk · 2025-06-25

**Clarity:** 3
**Significance:** 3
**Originality:** 3
**Rating:** 5
**Confidence:** 4

**Summary:**

This paper focuses on extending existing diffusion models for higher-resolution image generation in a training-free manner. Specifically, the authors propose ScaleDiff, a novel approach capable of scaling up the generation resolution of diffusion models based on both U-Net and DiT. The framework incorporates two key components: Neighborhood Patch Attention (NPA) and Latent Frequency Mixing (LFM). Experimental results demonstrate that the proposed ScaleDiff efficiently achieves higher-resolution image generation while exhibiting better performance.

**Questions:**

Please refer to the 'Weaknesses' section for details.

My primary unresolved concerns focus on the following aspects:
1. **More comprehensive and convincing evaluations**.
2. **Further polish on figures and presentations**.
3. **Some essential baseline methods**.
4. **More related literature expected to be discussed and compared**.

Since the overall quality of the paper is relatively impressive, I think I will be glad to raise my rating if the authors can address my concerns.

**Ethical Concerns:**

["NO or VERY MINOR ethics concerns only"]

**Final Justification:**

I have carefully read the other reviewers' comments and the authors' responses.
I believe the authors have addressed most of my concerns, and based on my pre-rebuttal assessment of this paper, I consider it to be a work of good overall quality, in terms of novelty, core contributions, and presentations.
Therefore, I have adjusted my final score accordingly. By the way, I hope the authors can further improve their paper based on all reviewers' suggestions, including discussions and comparisons of related work, as well as clearer presentation.
Additionally, I hope, in the revised version, the authors can complete those experiments that couldn't be finished within the limited rebuttal period, as they could further enhance the paper's overall quality and contribution.

**Limitations:**

Yes, the authors make an attempt to solve the existing limitations.

**Paper Formatting Concerns:**

None.

**Quality:**

3

**Strengths And Weaknesses:**

**Strengths**:

1. **Writing and Presentation**: The paper is clear, well-written, and easy to follow.
2. **Concise and Effective Design**: The proposed ScaleDiff is a training-free approach that can be broadly applied to existing diffusion models based on U-Net and DiT.
3. **Outstanding Performance**: Experimental results demonstrate that ScaleDiff can efficiently generate higher-resolution images of high quality, illustrating its effectiveness and efficiency.
4. **Necessary Ablation Studies**: The paper includes essential ablation studies to validate the effectiveness of the proposed key modules, making the results more convincing.
5. **Comprehensive Visualization**: The authors also provide additional qualitative results at various resolution scales in the supplementary material, further confirming the method's strong scalability and promising application potential.

**Weaknesses**:

1. **Limited Evaluation Scope**: The current quantitative evaluation is conducted on 1,000 samples from LAION, which is somewhat limited and may introduce performance variance. Thus, a larger-scale evaluation testbed is expected to make the results more robust and convincing.
2. **More Comprehensive Evaluation and Metrics**: While the authors provide synthesis results at other scales, quantitative results (similar to ScaleCrafter) at extreme aspect ratios (e.g., 4:1 and 1:4) could further highlight ScaleDiff’s universality. Moreover, beyond FID and patch-based FID, metrics like KID and patch-based KID are also widely adopted for evaluating higher-resolution image generation performance.
3. **Figures and Explanation**: I recommend refining Figure 3 to reduce blank space and improve layout efficiency. Additionally, it would be helpful to explicitly illustrate how low-frequency and high-frequency components are derived in the diagram (readers might intuitively assume bandpass filtering, while the method actually uses downsampling+upsampling). Furthermore, the hypothesis in Line 189 is confusing: why do imperfect images generated via latent-space upsampling provide high-quality high-frequency components? A deeper explanation, either theoretical or experimental, would help readers better understand this subsection.
4. **More Key Baselines**: The paper would benefit from including comparisons with more baseline approaches. For instance, replacing BSRGAN with StableSR [1*] or ESRGAN [2*], as well as conventional approaches like Bilinear/Bicubic interpolation, is expected to better demonstrate the advantages of the proposed ScalDiff.
5. **More Related Work to be discussed**: The paper would benefit from a more thorough discussion and comparison with recent advances in high-resolution image/video generation, such as URAE [3*], MegaFusion [4*], BeyondScene [5*], ResAdapter [6*], FreeScale [7*], and LinFusion [8*]. (especially those post-processing and tuning-free ones)

[1*] StableSR: Exploiting Diffusion Prior for Real-World Image Super-Resolution, in IJCV 2024.

[2*] ESRGAN: Enhanced Super-Resolution Generative Adversarial Networks, in ECCV workshop 2018.

[3*] Ultra-Resolution Adaptation with Ease, in ICML 2025.

[4*] MegaFusion: Extend Diffusion Models towards Higher-resolution Image Generation without Further Tuning, in WACV 2025.

[5*] BeyondScene: Higher-Resolution Human-Centric Scene Generation With Pretrained Diffusion, in ECCV 2024.

[6*] ResAdapter: Domain Consistent Resolution Adapter for Diffusion Models, in AAAI 2025.

[7*] FreeScale: Unleashing the Resolution of Diffusion Models via Tuning-Free Scale Fusion, in arXiv 2024.

[8*] LinFusion: 1 GPU, 1 Minute, 16K Image, in arXiv 2024.

---

> ### Author Rebuttal · Authors · 2025-07-31
>
> Thank you for your comprehensive and constructive feedback. We are grateful for your positive assessment, particularly your recognition of our method's effective design and outstanding performance. In response to your valuable suggestions, we have conducted additional experiments to provide a more comprehensive evaluation against new metrics and baselines. We hope that these results and our detailed responses fully address your concerns.
>
> >**W1: Limited Evaluation Scope**
>
> We acknowledge that evaluating with 1,000 samples has its limitations. However, for fair comparison with \[1,2,3\],  we use 1,000 samples to follow the common practice established by the prior works.
>
> >**W2: More Comprehensive Evaluation and Metrics**
>
> Thank you for the constructive feedback. In response to your suggestion, we have conducted additional experiments to include KID and patch-based KID metrics (KID\_p). The results are as follows:
>
> | Model | Resolution | Method | KID↓ | KID\_p↓ |
> | ----- | ----- | :---- | :---- | :---- |
> | SDXL | 2048 | SDXL Direct | 0.0124 | 0.0137 |
> |  |  | SDXL \+ BSRGAN | 0.0041 | 0.0092 |
> |  |  | UltraPixel | 0.0056 | 0.0093 |
> |  |  | ScaleCrafter | 0.0033 | 0.0064 |
> |  |  | HiDiffusion | 0.0040 | 0.0067 |
> |  |  | DiffuseHigh | 0.0033 | 0.0062 |
> |  |  | DemoFusion | 0.0032 | **0.0050** |
> |  |  | ScaleDiff (Ours) | **0.0032** | 0.0067 |
> |  |  |  |  |  |
> |  | 4096 | SDXL Direct | 0.0717 | 0.0250 |
> |  |  | SDXL \+ BSRGAN | 0.0044 | 0.0160 |
> |  |  | UltraPixel | 0.0055 | 0.0112 |
> |  |  | ScaleCrafter | 0.0110 | 0.0217 |
> |  |  | HiDiffusion | 0.0216 | 0.0494 |
> |  |  | DiffuseHigh | 0.0034 | 0.0079 |
> |  |  | DemoFusion | 0.0041 | **0.0079** |
> |  |  | ScaleDiff (Ours) | **0.0025** | 0.0080 |
> |  |  |  |  |  |
> | FLUX | 2048 | FLUX Direct | 0.0069 | 0.0086 |
> |  |  | FLUX \+ BSRGAN | 0.0052 | 0.0081 |
> |  |  | ScaleDiff (Ours) | **0.0047** | **0.0073** |
> |  |  |  |  |  |
> |  | 4096 | FLUX Direct | 0.2775 | 0.2642 |
> |  |  | FLUX \+ BSRGAN | 0.0051 | 0.0125 |
> |  |  | ScaleDiff (Ours) | **0.0044** | **0.0098** |
>
> Our ScaleDiff demonstrated favorable performance on the newly added KID and KID\_p metrics as well.
> Due to the limited time for the rebuttal, we will include the evaluation on extreme aspect ratios in the final version of the paper.
>
> >**W3: Figures and Explanation**
>
> Thank you for these concrete suggestions. We will revise Figure 3 to be more space-efficient and to explicitly illustrate that the frequency components are derived via downsampling-upsampling, not bandpass filtering.
>
> Regarding the LFM hypothesis,  we provide both a detailed analysis in Appendix C and new quantitative evidence from an experiment designed to justify the hypothesis. In the new experiment, we compare three upsampling methods: (1) Latent-space upsampling (Z\_latent), (2) RGB-space upsampling (Z\_rgb), and (3) our proposed LFM. The detailed experimental setting can be found in our response to Reviewer 2Ebs (Q4). We measure PSNR for fidelity and Image Entropy and Mean Variance of Laplacian (mVoL) to evaluate sharpness and the degree of oversmoothing.
> The results are as follows:
>
> |  | PSNR↑ | Entropy↑ | mVoL↑ |
> | :---- | :---- | :---- | :---- |
> | Z\_latent | 20.14 | 7.12 | **525.78** |
> | Z\_rgb | **23.58** | 7.07 | 49.69 |
> | **LFM (Ours)** | 21.42 | **7.16** | 382.67 |
>
> Z\_rgb achieves a higher PSNR, but its extremely low Entropy and mVoL scores quantitatively demonstrate the oversmoothing problem. In contrast,  Z\_latent, despite having a lower PSNR due to artifacts, achieves significantly higher Entropy and mVoL scores, confirming it effectively mitigates oversmoothing.
> LFM leverages  Z\_latent as a base to avoid oversmoothing, while replacing its high-frequency components with those from Z\_rgb to resolve the artifact issue.
> We will add this table and a clearer explanation to the final manuscript.
>
> >**W4: More Key Baselines**
>
> Thank you for the suggestion. We have conducted additional comparisons with ESRGAN and Bicubic interpolation:
>
> | Model | Resolution | Method | FID↓ | KID↓ | IS↑ | FID\_p↓ | KID\_p↓ | IS\_p↑ | CLIP↑ |
> | ----- | ----- | :---- | :---- | :---- | :---- | :---- | :---- | :---- | :---- |
> | SDXL | 2048 | ESRGAN | 65.38 | 0.0048 | 17.83 | 41.97 | 0.0093 | 22.01 | 33.01 |
> |  |  | Bicubic | 64.20 | 0.0039 | 18.84 | 49.08 | 0.0162 | 21.52 | 33.05 |
> |  |  | ScaleDiff (Ours) | **62.98** | **0.0032** | **19.54** | **38.03** | **0.0067** | **25.70** | **33.11** |
> |  |  |  |  |  |  |  |  |  |  |
> |  | 4096 | ESRGAN | 65.43 | 0.0048 | 17.65 | 49.31 | 0.0158 | 16.63 | 33.01 |
> |  |  | Bicubic | 64.20 | 0.0039 | 18.88 | 50.31 | 0.0161 | 18.23 | **33.05** |
> |  |  | ScaleDiff (Ours) | **61.87** | **0.0025** | **19.56** | **38.89** | **0.0080** | **20.41** | 33.04 |
> |  |  |  |  |  |  |  |  |  |  |
> | FLUX | 2048 | ESRGAN | 65.01 | 0.0053 | 18.87 | 41.32 | 0.0078 | 23.14 | 31.16 |
> |  |  | Bicubic | 64.56 | 0.0054 | **19.41** | 42.69 | 0.0101 | 22.64 | **31.26** |
> |  |  | ScaleDiff (Ours) | **64.31** | **0.0047** | 18.51 | **40.03** | **0.0073** | **23.38** | 31.22 |
> |  |  |  |  |  |  |  |  |  |  |
> |  | 4096 | ESRGAN | 65.01 | 0.0053 | 18.87 | 48.88 | 0.0119 | 17.29 | 31.15 |
> |  |  | Bicubic | 64.56 | 0.0054 | **19.30** | 54.24 | 0.0190 | 16.84 | **31.26** |
> |  |  | ScaleDiff (Ours) | **64.06** | **0.0044** | 18.36 | **44.29** | **0.0098** | **17.41** | 31.14 |
>
> As shown in the table, our method provides a clear advantage over these baselines as well. Traditional upscalers like ESRGAN and Bicubic often fall short in generating the fine details. This is strongly reflected in the patch-level fidelity metrics.
>
> >**W5: More Related Work to be discussed**
>
> Thank you for providing a list of relevant recent works. We will expand our related work section in the final manuscript to thoroughly discuss these advances.
> For instance, FreeScale enhances overall structure and detail via Scale Fusion, which mixes frequencies from global and local attention. However, its local attention calculation introduces computational redundancy, an issue our NPA is specifically designed to solve. Furthermore, FreeScale's reliance on dilated convolution makes it incompatible with DiT architectures.
> MegaFusion employs a truncate and relay strategy that continues the denoising process at a higher resolution from an intermediate low-resolution step. This differs from our upsample-diffuse-denoise pipeline, requiring fewer total denoising steps. However, MegaFusion is not demonstrated at very high resolutions like 4096x4096
> For a quantitative comparison with some of these recent methods, please refer to our response to Reviewer e7Kz (W2).
>
> \[1\] DemoFusion: Democratising High-Resolution Image Generation With No $$$, in CVPR 2024
> \[2\] Is One GPU Enough? Pushing Image Generation at Higher-Resolutions with Foundation Models, in NeurIPS 2024
> \[3\] DiffuseHigh: Training-free Progressive High-Resolution Image Synthesis through Structure Guidance, in AAAI 2025

---

> > ### Comment · Reviewer_YvLk · 2025-08-02
> >
> > Thanks for the authors' effort and responses.
> > I have carefully read the other reviewers' comments and the authors' responses.
> > I believe the authors have addressed most of my concerns; thus, I have adjusted my final score accordingly.
> > By the way, I hope the authors can further improve their paper based on all reviewers' suggestions, including discussions and comparisons of related work, as well as clearer presentation.
> > Additionally, I hope, in the revised version, the authors can complete those experiments that couldn't be finished within the limited rebuttal period, as they could further enhance the paper's overall quality and contribution.

---

> ### Author Response · Authors · 2025-08-03
> **Thank you!**
>
> We thank you for your constructive review and for the positive reassessment of our manuscript.
> Your valuable recommendations have been noted. We will ensure these improvements are incorporated into the camera-ready version upon the paper's acceptance.

---

### Official Review · Reviewer_YkCW · 2025-06-29

**Clarity:** 3
**Significance:** 2
**Originality:** 2
**Rating:** 4
**Confidence:** 3

**Summary:**

This paper presents ScaleDiff, a training-free, resolution-agnostic framework for extending the generation resolution of pretrained text-to-image diffusion models, including both U-Net and Diffusion Transformer architectures. The method introduces Neighborhood Patch Attention (NPA) to reduce redundancy in self-attention over non-overlapping patches, significantly improving efficiency over standard patch-based approaches. Additionally, Latent Frequency Mixing (LFM) enhances fine detail synthesis, while Structure Guidance improves global coherence during denoising. Integrated into an upsample-diffuse-denoise pipeline, ScaleDiff achieves state-of-the-art performance in image quality and inference speed compared to other training-free high-resolution generation methods.

**Questions:**

+ Include some analysis or visual examples of where ScaleDiff fails or degrades (e.g., extreme compositions, long prompts, or cluttered scenes). How does the system behave under such inputs?

**Ethical Concerns:**

["NO or VERY MINOR ethics concerns only"]

**Final Justification:**

The author address my concerns well but I still believe this work is not strong enough to guarantee a score of **5** so I keep original score of **4**.

**Limitations:**

No other limitations

**Paper Formatting Concerns:**

No paper formatting concerns

**Quality:**

3

**Strengths And Weaknesses:**

**Strength**

+ Works with both U-Net and Diffusion Transformer backbones, broadening applicability beyond methods tailored only for U-Nets.

+ The proposed Neighborhood Patch Attention (NPA) effectively reduces redundant computations while preserving receptive field coverage, improving speed and scalability.

+ Significant improvements in FID and CLIP scores on multiple datasets and backbones. Visuals show noticeably better high-frequency details.

**Weaknesses**

+ The porposed techniques (NPA, LFM, SG), while efficient in theory, may require non-trivial engineering to integrate into existing pipelines as they looks quite complex to implement, especially NPA.

+ Lack of discussion about failure cases.

---

> ### Author Rebuttal · Authors · 2025-07-31
>
> We are sincerely grateful for your valuable feedback and positive evaluation. We are encouraged that you recognized our method's broad applicability across different architectures and the efficiency of our Neighborhood Patch Attention (NPA). We have carefully addressed the concerns you raised in our detailed response below and hope it fully clarifies the points you raised.
>
> >**W1: Implementation complexity of proposed techniques.**
>
> We understand this concern. To facilitate adoption and verify our implementation's simplicity, we will release our code as open-source as well as the supplementary material upon publication. We are confident that this will demonstrate the ease of integrating ScaleDiff into existing diffusion model frameworks.
>
> >**W2/Q1: Discussion of failure cases.**
>
> Thank you for the suggestion. We have included a "Limitations" section in Appendix E, which discusses some general failure cases, such as inconsistent local content in close-ups and repetitive background textures. In response to your suggestion, we have also experimented with the examples you mentioned (extreme compositions, long prompts, cluttered scenes). Our analysis shows that since the global structure of the final image is guided by the initial image generated at the native resolution, the performance in these challenging scenarios is largely dependent on the capabilities of the base model itself. We will add a new figure to the final version illustrating these results to provide a more comprehensive understanding of our method's limitations.

---

> ### Author Response · Authors · 2025-08-05
>
> Dear Reviewer ykCW,
>
> Thank you once again for your thoughtful and constructive review. As the discussion deadline approaches, we wanted to kindly check if you had any additional questions or concerns we could clarify. We truly appreciate your time and effort during this busy period and would be happy to provide further details on any aspect of our work.
>
> Best regards,
> Authors

---

> ### Comment · Reviewer_YkCW · 2025-08-06
>
> The author address my concerns well but I still believe this work is not strong enough to guarantee a score of **5** so I keep original score of **4**.

---

### Official Review · Reviewer_e7Kz · 2025-07-01

**Clarity:** 3
**Significance:** 3
**Originality:** 3
**Rating:** 4
**Confidence:** 4

**Summary:**

To address the degradation in image quality when existing open-source text-to-image models generate outputs beyond their training resolution, this paper proposes ScaleDiff, a training-free framework composed of three components: NPA, LFM, and SG. Specifically, NPA eliminates the additional computational overhead introduced by previous patch-based methods, LFM mitigates detail loss caused by upsampling in pixel space, and SG enhances global consistency in the generated images. Qualitative and quantitative experiments validate the effectiveness of the method.

**Questions:**

I will consider increasing the score once all concerns are addressed.

**Ethical Concerns:**

["NO or VERY MINOR ethics concerns only"]

**Final Justification:**

Most of my concerns have been addressed, except for some clarifications, such as the explanation of why the previous method is difficult to adapt to DiT and the explanation of LFM for helping preserve details. Consequently, I have raised my rating to borderline accept.

**Limitations:**

yes

**Quality:**

3

**Strengths And Weaknesses:**

Strengths:
1. The paper is well-written with a clear motivation, making it easy to understand.
2. The proposed NPA is interesting; it is a training-free and model-agnostic attention mechanism that allows effective aggregation of contextual information for each token without introducing additional computational overhead.
3. The proposed pipeline demonstrates its effectiveness by outperforming several previous approaches when applied to the pretrained SDXL model.

Weaknesses:
1. The paper claims that previous methods are not applicable to diffusion models based on the DiT architecture, but it does not provide adequate qualitative or quantitative evidence to support this claim. I noticed that Figure 1 displays a qualitative comparison of MultiDiffusion applied to DiT, yet MultiDiffusion is not a recent state-of-the-art method.
2. While the paper includes comparisons with earlier methods, it would benefit from evaluations against more recent models such as AccDiffusion v2, FreeScale, AP-LDM, and MegaFusion.
3. Although the proposed method is effective, it lacks in-depth explanations of certain techniques. For instance, why does interpolating high-frequency information in the latent space with LFM help preserve details?
4. Line 46 of the paper mentions that upsampling in RGB space may lead to over-smoothing, which is indeed a valid concern. However, to the best of my knowledge, most existing methods that perform RGB-space upsampling typically apply a diffusion model for refinement afterward. Therefore, I am concerned that the issue raised by the paper may not be particularly significant.
5. I noticed that using SG alone may have a negative impact on the model’s output, both in qualitative and quantitative experiments. This raises concerns that the generalizability of SG might be limited.
6. According to previous studies [1], directly changing the sequence length of the attention inputs in a training-free setting can lead to degraded attention performance. How does NPA address the risk of such degradation?

[1] Training-free Diffusion Model Adaptation for Variable-Sized Text-to-Image Synthesis. NIPS2023.

---

> ### Author Rebuttal · Authors · 2025-07-31
>
> We are sincerely grateful for your detailed and insightful feedback. We appreciate your recognition of our paper's clear motivation and the novelty of the proposed Neighborhood Patch Attention (NPA). To address your main concerns, we have conducted additional experiments and provided in-depth clarifications for our design choices. We hope these additions and clarifications fully address your concerns.
>
> >**W1: Lack of evidence for inapplicability to DiT**
>
> The difficulty in applying existing training-free methods to DiT architectures stems from their foundational designs, which often make them ineffective. Existing training-free methods can be broadly categorized into three groups:
>
> * **Parameter Rectification-based methods** (e.g., ScaleCrafter) typically adjust convolution layers during inference. These methods are fundamentally inapplicable to DiT models without convolution layers.
> * **SDEdit-based methods** (e.g., Diffusehigh) rely on the base model's ability to generate high-frequency information at high resolutions. As shown in Figure 1, while U-Net models often retain this capability, DiT-based models do not. Accordingly, as can be seen in Figure 1, DiT fails to generate local details when the SDEdit pipeline is applied.
> * **Patch-based methods** (e.g., DemoFusion) are model-agnostic as they operate by dividing a high-resolution canvas into native-resolution patches for individual processing. This approach, however, introduces significant computational redundancy from handling overlapping regions.
>
> This leaves patch-based methods as the most applicable option for DiT, yet they suffer from significant computational bottlenecks, a limitation our work is designed to overcome.
> We will revise the final version upon acceptance to clearly demonstrate these limitations and add a new figure showing the results of applying prior methods directly to the FLUX (DiT) model.
>
> >**W2: Comparison with more recent SOTA models.**
>
> In response to your feedback, we have conducted additional experiments comparing ScaleDiff with recent methods, including AccDiffusion v2 (TPAMI 2025\), FreeScale (ICCV 2025), AP-LDM (arXiv 2024), and MegaFusion (WACV2025) at 2048x2048 resolution.
>
> |  | FID↓ | KID↓ | IS↑ | FID\_p↓ | KID\_p↓ | IS\_p↑ | CLIP↑ | Time(s) |
> | :---- | :---- | :---- | :---- | :---- | :---- | :---- | :---- | :---- |
> | AccDiffusion v2 | 64.28 | 0.0039 | 18.83 | 39.47 | 0.0075 | 25.82 | 33.03 | 207 |
> | FreeScale | 63.78 | **0.0029** | 18.86 | **37.85** | **0.0060** | 24.65 | 33.08 | 69 |
> | AP-LDM | 63.18 | 0.0033 | 17.40 | 38.99 | 0.0075 | 23.81 | 32.90 | 37 |
> | MegaFusion | 65.42 | 0.0041 | 17.92 | 38.45 | 0.0063 | 23.09 | 33.05 | 30 |
> | ScaleDiff (Ours) | **62.98** | 0.0032 | **19.54** | 38.03 | 0.0067 | **24.70** | **33.11** | 31 |
>
> Results show that our method outperforms recent approaches like AccDiffusion v2, AP-LDM, and MegaFusion.
> In addition, our method shows highly competitive results against FreeScale and notably surpasses it in efficiency with a more than 2x faster inference time.
> Due to the time constraints of the rebuttal period, we will include the full comparison at 4096x4096 resolution as well in the final version to provide a more comprehensive evaluation.
>
> >**W3: In-depth explanation of LFM preserving details.**
>
> We provide a detailed analysis and motivation for our Latent Frequency Mixing (LFM) technique in the supplementary material (Appendix C). Our analysis shows that while latent-space upsampling (Z\_latent) avoids oversmoothing, it lacks the high-frequency components necessary to prevent artifacts when decoded. Conversely, simply using RGB-space upsampled latent leads to an oversmoothing bias in the final result. Therefore, LFM leverages the low-frequency components from Z\_latent as a base to avoid oversmoothing, while replacing its high-frequency components with those from Z\_rgb to resolve the artifact issue. We will clarify this explanation in the main paper in the final version upon acceptance to improve readability.
>
> >**W4: Significance of RGB-space upsampling oversmoothing issue.**
>
> The core goal of our research is to address the oversmoothing that occurs in the final results **after refining** an RGB-upsampled image with a diffusion model. This is a known challenge that has been previously identified in works like DiffuseHigh \[1\] and LSRNA \[2\]. Our work tackles this specific problem by introducing Latent Frequency Mixing (LFM), which provides a better-balanced reference latent that is less biased towards a smooth solution. Our ablation study (Table 3, Figure 5b vs 5c) empirically supports this, showing significant improvements in patch-level detail metrics (FIDp​, ISp​) when LFM is applied.
>
> >**W5: Negative impact and limited generalizability of SG.**
>
> As we state in our ablation study, applying SG results in a clear mitigation of object repetition. We have demonstrated this in Figure 5(c), which shows a marked reduction in such artifacts compared to the baseline without SG in Figure 5(a). While the quantitative results in Table 3 show a slight drop in one metric (IS) when comparing the baseline (a) to the SG-only version (c), we would like to highlight that SG improves in FID, FID\_p, IS\_p, and the CLIP score, demonstrating its positive effect on image fidelity and text alignment. ​Most importantly, our results in Table 3 show that the complete ScaleDiff pipeline (LFM+SG) achieves the best overall performance, outperforming the LFM-only version (b). This demonstrates that SG is an indispensable component for improving global structure.
>
>
> >**W6: How NPA avoids attention degradation from sequence length changes.**
>
> Our proposed method does not suffer from the performance degradation issue mentioned in \[3\]. The problem described in \[3\] arises from an increase in the number of key/value tokens during the attention calculation at higher resolutions. However, our NPA is designed to maintain the number of key and value tokens for each query to be identical to that of the native resolution. Since the number of tokens per attention operation does not change, the performance degradation is effectively avoided.
>
> \[1\] DiffuseHigh: Training-free Progressive High-Resolution Image Synthesis through Structure Guidance, in AAAI 2025
> \[2\] Latent Space Super-Resolution for Higher-Resolution Image Generation with Diffusion Models, in CVPR 2025
> \[3\] Training-free Diffusion Model Adaptation for Variable-Sized Text-to-Image Synthesis, in NeurIPS 2023\.

---

> ### Author Response · Authors · 2025-08-05
>
> Dear Reviewer e7kz,
>
> Thank you once again for your thoughtful and constructive review. As the discussion deadline approaches, we wanted to kindly check if you had any additional questions or concerns we could clarify. We truly appreciate your time and effort during this busy period and would be happy to provide further details on any aspect of our work.
>
> Best regards,
> Authors

---

> ### Comment · Reviewer_e7Kz · 2025-08-07
> **Reply to the Rebuttal**
>
> Thanks for the rebuttal and additional experiments. Most of my concerns have been addressed. I've decided to raise my score and hope the authors can improve the revised version based on the rebuttal.

---

### Official Review · Reviewer_2Ebs · 2025-07-02

**Clarity:** 2
**Significance:** 4
**Originality:** 3
**Rating:** 5
**Confidence:** 4

**Summary:**

The authors propose a method for extending the resolution of the output images from pretrained diffusion models without additional training. For this, the authors introduce Neighborhood Patch Attention. This helps in reducing computational redundancy in the self-attention layer with non-overlapping patches. They also introduce Latent Frequency Mixing for generating fine-grained details in the output images. Additionally, global consistency is enforced through Structure Guidance, applied in the latent space.

**Questions:**

1. In the context of upscaling pipeline, how is the latent downsampled and upsampled? Please provide a detailed description of how to compute the high-frequency components.
2.  The shifted crop sampling strategy should be clearly explained.
3. The authors need to discuss whether there is a possibility of information loss due to shifted crop sampling. If so, how does the proposed method deal with this?
3. The authors hypothesize that upscaling the latent in the latent space can mitigate oversmoothing. The basis of this hypothesis needs to be explained.
4. The process of creating $Z_{latent}$ and the motivation of using it should be explained.

**Ethical Concerns:**

["NO or VERY MINOR ethics concerns only"]

**Final Justification:**

My concerns are resolved.

**Limitations:**

Yes.

**Paper Formatting Concerns:**

None.

**Quality:**

3

**Strengths And Weaknesses:**

Strength
1. The ideas of avoiding patch-based processing for non-self-attention layers and extracting queries from non-overlapping patches to reduce computation, and latent frequency mixing for image quality enhancement are interesting.
2. The proposed method can be integrated with pretrained diffusion models without additional training.
3. The proposed method shows promising results in terms of processing time and image quality.


Weakness
1. In the context of upscaling pipeline, the process of computing the high-frequency components is ambiguous. How is the latent downsampled and upsampled in this context?
2.  The shifted crop sampling strategy is not clearly explained. The authors have also not discussed whether there is a possibility of information loss due to shifted crop sampling. If so, how does the proposed method deal with this potential problem?
3. The authors hypothesize that upscaling the latent in the latent space can mitigate oversmoothing. However, the basis of this hypothesis is not described in the paper.
4. The process of creating $Z_{latent}$ is not clear to me. Does it consist of components from both the RGB space and the latent space? If so, how are these two components blended? What is the motivation behind it?

---

> ### Author Rebuttal · Authors · 2025-07-31
>
> We are sincerely grateful for your positive and constructive review. Thank you for recognizing the key strengths of our work, particularly the novelty of our Neighborhood Patch Attention (NPA) for efficiency and Latent Frequency Mixing (LFM) for enhancing detail. We have addressed all of your questions below with detailed explanations and new experimental evidence, and we hope these clarifications fully resolve your concerns.
>
> >**Q1: Latent up/downsampling & high-frequency component computation.**
>
> As briefly mentioned in Lines 196-198, we use bicubic interpolation for all downsampling and upsampling operations. To compute the frequency components, we follow the approach from prior work \[1\]. Specifically, the low-frequency component of a latent (Z) is obtained by downsampling it and then upsampling it back to the original size (i.e., upsample(downsample(Z))). The high-frequency component is then derived by subtracting this low-frequency component from the original latent (Z−Z low ). We acknowledge that this could be stated more clearly and will revise Section 3.3 in the final version to make this process more explicit.
>
> >**Q2: Explanation of shifted crop sampling.**
>
> Algorithm 1 in our supplementary material shows the detailed patch extraction procedure for our method. To explain it more intuitively, shifted crop sampling is a patch extraction method analogous to a standard convolution operation; we use a sliding window that moves from the top-left corner of the feature map to crop patches. We acknowledge that this explanation was insufficient in the main text and will revise it in the final version with a direct reference to Algorithm 1\.
>
> >**Q3: Information loss from shifted crop sampling.**
>
> We agree that each patch in patch-by-patch image processing lacks global context and, therefore, can lead to the generation of repetitive objects across the image.
> In Figure 1, we have also found these repetitive objects in the existing method (MultiDiffusion).
> Our ScaleDiff pipeline is specifically designed to address this information loss by using a native resolution reference image for global guidance and using NPA  for efficient local synthesis.
> Furthermore, as shown in Appendix Table 1,  our proposed NPA outperforms the standard attention mechanism (Base) when integrated into the ScaleDiff Pipeline, confirming the effectiveness of our approach.
>
> >**Q4: Basis for the hypothesis on latent-space upscaling mitigates oversmoothing.**
>
> To provide quantitative evidence for our approach, we performed an additional experiment, which will be included in the final paper. The process is as follows:
>
> 1. Generate 1K 1024x1024 images to use as a Ground Truth,
> 2. Downsample these images to 512x512, then apply the upsample-diffuse-denoise pipeline to restore them to 1024x1024.
> 3. We compare three methods for upsampling: (1) Latent-space upsampling (Z\_latent), (2) RGB-space upsampling (Z\_rgb), and (3) our proposed LFM.
> 4. To evaluate the results, we measure PSNR for fidelity, and Image Entropy and Mean Variance of Laplacian (mVoL) to further evaluate the degree of sharpness.
>
> |  | PSNR↑ | Entropy↑ | mVoL↑ |
> | :---- | :---- | :---- | :---- |
> | Z\_latent | 20.14 | 7.12 | **525.78** |
> | Z\_rgb | **23.58** | 7.07 | 49.69 |
> | **LFM (Ours)** | 21.42 | **7.16** | 382.67 |
>
> Our results show that Z\_latent achieves better Image Entropy and mVoL compared to Z\_rgb, which demonstrates Z\_latent mitigates oversmoothing of Z\_rgb. On the other hand, Z\_rgb shows better PSNR, representing lower artifacts than Z\_latent. Our LFM leverages this benefit from both latent features by combining them and shows favorable scores on all the metrics. We will add a table with this analysis to the final version.
>
> >**Q5: The process of creating Z\_latent and the motivation for using it should be explained.**
>
> Following prior work \[2\], we create Z\_latent​  by performing bicubic interpolation on the low-resolution latent Z\_L​  directly within the latent space.
> The motivation for blending Z\_latent and Z\_rgb is to combine their complementary characteristics, as analyzed in Appendix C. Our analysis shows that while Z\_latent avoids an oversmoothing bias, its weak high-frequency component leads to artifacts when decoded. Therefore, LFM leverages the low-frequency components from Z\_latent as a base to avoid oversmoothing, while replacing its deficient high-frequency components with those from Z\_rgb to resolve the artifact issue.
> We acknowledge that this explanation could be stated more clearly in the main paper. We will revise the manuscript to ensure this mechanism is easier to follow in the final version.
>
> \[1\] FreCaS: Efficient Higher-Resolution Image Generation via Frequency-aware Cascaded Sampling, in ICLR 2025
> \[2\] DemoFusion: Democratising High-Resolution Image Generation With No $$$, in CVPR 2024

---

> > ### Comment · Reviewer_2Ebs · 2025-08-03
> >
> > The authors have addressed my comments in a detailed manner. The responses are mostly satisfactory. If possible, the authors should consider adding a theoretical analysis related to Q4 alongside the experimental results.

---

> ### Author Response · Authors · 2025-08-05
>
> Thank you for your valuable feedback and suggestions regarding a more theoretical analysis of the oversmoothing issue.
> As discussed in works like DiffuseHigh [1], we conjecture that the oversmoothing arises because RGB space upsampling creates an image close to the "blurry data distribution mode," which also guides the model generation in the same mode.
> While methods like DiffuseHigh use explicit sharpening to shift this image, we believe our latent space upsampling implicitly shifts the image distribution, resulting in a more realistic reference image.
> The effectiveness of our approach in generating a sharp reference image is also demonstrated in our quantitative experiment in the rebuttal in terms of mVoL score.
> We will incorporate this discussion into the final manuscript as well. Please let us know if there is anything we can further clarify or discuss for the positive reassessment of our paper.
>
> [1] DiffuseHigh: Training-free Progressive High-Resolution Image Synthesis through Structure Guidance, in AAAI 2025

---

> > ### Comment · Reviewer_2Ebs · 2025-08-05
> >
> > Thanks for the response. Consider adding this in the manuscript.

---

### Note · Authors · 2025-08-13

We sincerely thank all reviewers for their time and valuable insights. We are encouraged by the positive feedback and have addressed all major concerns, as summarized below.

### **Positive Feedback and Constructive Discussion**

Reviewers reached a consensus on the core strengths of our work, noting:

* **The novelty and efficiency of our Neighborhood Patch Attention (NPA)** for reducing computational overhead in a training-free manner (2Ebs, e7Kz, YkCW).
* **The effective model-agnostic design of ScaleDiff**, applicable to both U-Net and Diffusion Transformer architectures (e7Kz, YkCW, YvLk).
* **Strong performance in generating high-quality images** with significant improvements over prior methods (2Ebs, e7Kz, YkCW, YvLk).

Our rebuttal and additional experiments—comparing against recent SOTA models and expanding metrics—successfully resolved most initial concerns. Several reviewers raised their scores after confirming their concerns were addressed (e7Kz, YvLk), while others expressed satisfaction with our responses (2Ebs, YkCW).

### **Commitment to the Final Version**

We will incorporate all feedback into the camera-ready version by:

* **Strengthening Experimental Validation:** Adding comparisons with the latest SOTA models, broader metrics (KID, extreme aspect ratios), and a full τ ablation (e7Kz, YvLk, BUQE).
* **Improving Clarity and Presentation:** Refining figures, adding controlled comparisons and DiT results, expanding theoretical discussion, and clarifying technical details (YvLk, BUQE, 2Ebs).
* **Expanding Discussion and Reproducibility:** Adding failure case analysis, updating Related Work and Limitations, and releasing code for full reproducibility (YvLk, YkCW).

We appreciate the constructive engagement, which has strengthened our paper, and look forward to presenting an improved final version.

Best regards,
Authors

---

### Decision · Program_Chairs · 2025-09-17

**Decision:**

Accept (poster)

**Comment:**

ScaleDiff proposes a training-free, model-agnostic framework that effectively extends the resolution of diffusion models across both U-Net and DiT architectures. Its key components—Neighborhood Patch Attention, Latent Frequency Mixing, and Structure Guidance—are well-motivated, and shown through extensive experiments to yield competitive or superior results compared to recent baselines while remaining efficient. While some concerns remain about presentation clarity and completeness of comparisons, the reviewers generally agree that the core contributions represent a meaningful advancement for high-resolution image synthesis.